# Benchmarking Offline Reinforcement Learning in Factorisable Action Spaces

## Abstract

Extending reinforcement learning (RL) to offline contexts is a promising prospect, particularly in sectors where data collection poses substantial challenges or risks. Pivotal to the success of transferring RL offline is mitigating overestimation bias in value estimates for state-action pairs absent from data. Whilst numerous approaches have been proposed in recent years, these tend to focus primarily on continuous or small-scale discrete action spaces. Factorised discrete action spaces, on the other hand, have received relatively little attention, despite many real-world problems naturally having factorisable actions. In this work, we undertake an initial formative investigation into offline reinforcement learning in factorisable action spaces. Using value-decomposition as formulated in DecQN as a foundation, we conduct an extensive empirical evaluation of several offline techniques adapted to the factorised setting. In the absence of established benchmarks, we introduce a suite of our own based on a discretised variant of the DeepMind Control Suite, comprising datasets of varying quality and task complexity. Advocating for reproducible research and innovation, we make all datasets available for public use, alongside our code base.

## 1 Introduction

The idea of transferring the successes of reinforcement learning (RL) to the offline setting is an enticing one. The opportunity for agents to learn optimal behaviour from sub-optimal data prior to environment interaction extends RL's applicability to domains where data collection is costly, time-consuming, or dangerous (Lange et al., 2012). This includes not only those domains where RL has traditionally found favour, such as games and robotics (Mnih et al., 2013; Hessel et al., 2018; Kalashnikov et al., 2018; Mahmood et al., 2018), but also areas in which online learning presents significant practical and/or ethical challenges, such as autonomous driving (Kiran et al., 2022) and healthcare (Yu et al., 2021a).

Unfortunately, taking RL offline is not as simple as naively applying standard off-policy algorithms to pre-existing datasets and removing online interaction. A substantial challenge arises from the compounding and propagation of overestimation bias in value estimates for actions absent from data (Fujimoto et al., 2019b). This bias stems from the underlying bootstrapping procedure used to derive such estimates and subsequent maximisation to obtain policies, whether implicit such as in Q-learning or explicit as per actor-critic methods (Levine et al., 2020). Fundamentally, agents find it difficult to accurately determine the value of actions not previously encountered, and thus any attempt to determine optimal behaviour based on these values is destined to fail. In online learning, such inaccuracies can be compensated for through continual assimilation of environmental feedback, but offline such a corrective mechanism is no longer available.

In response to these challenges, there has been a plethora of approaches put forward that aim to both curb the detrimental effects of overestimation bias as well as let agents discover optimal policies, or at least improve over the policy/policies that collected the data to begin with (Levine et al., 2020). The last few years in particular have seen a wide variety of approaches proposed, making use of policy constraints (Fujimoto et al., 2019b; Zhou et al., 2021; Wu et al., 2019; Kumar et al., 2019; Kostrikov et al., 2021b; Fujimoto & Gu, 2021), conservative value estimation (Kostrikov et al., 2021a; Kumar et al., 2020), uncertainty estimation (An et al., 2021; Ghasemipour et al., 2022; Bai et al., 2022; Yang et al., 2022; Nikulin et al., 2023; Beeson & Montana,

2024) and environment modelling (Kidambi et al., 2020; Yu et al., 2021b; Argenson & Dulac-Arnold, 2020; Janner et al., 2022), to name just a few. Each approach comes with its own strengths and weaknesses in terms of performance, computational efficiency, ease of implementation and hyperparameter tuning.

To date, most published research in offline-RL has focused on either continuous or small-scale discrete action spaces. However, many complex real-world problems can be naturally expressed in terms of *factorised action spaces*, where global actions consist of multiple distinct sub-actions, each representing a key aspect of the decision process. Examples include ride-sharing (Lin et al., 2018), recommender systems (Zhao et al., 2018), robotic assembly (Driess et al., 2020) and healthcare (Liu et al., 2020).

Formally, the factorised action space is considered the Cartesian product of a finite number of discrete sub-action spaces, *i.e.* $\mathcal{A} = \mathcal{A}_1 \times ... \times \mathcal{A}_N$, where $\mathcal{A}_i$ contains $n_i$ (sub-)actions and $N$ corresponds to the dimensionality of the action space. The total number of actions, often referred to as atomic actions, is thus $\prod_{i=1}^{N} n_i$, which can undergo combinatorial explosion if $N$ and/or $n_i$ grow large.

In recognition of this propsect, in online-RL various strategies have been devised to preserve the effectiveness of commonly used discrete algorithms (Tavakoli et al., 2018; Tang & Agrawal, 2020). The concept of *value-decomposition* (Seyde et al., 2022) is particularly prominent, wherein value estimates for each sub-action space are computed independently, yet are trained to ensure that their aggregate mean converges towards a universal value. This paradigm is inspired by the *centralised training and decentralised execution* framework popular in multi-agent RL (MARL) (Kraemer & Banerjee, 2016), in which each sub-action space is treated analogously to individual agents. The overall effect is to reduce the total number of actions for which a value needs to be learnt from a product to a sum, making problems with factorisable actions spaces much more tractable for approaches such as Q-learning.

In this work, we undertake an initial investigation into offline-RL for factorisable action spaces. Using value-decomposition, we show how a factorised approach provides several benefits over standard atomic representation. We conduct an extensive empirical evaluation of a number of offline approaches adapted to the factorised action setting, comparing the performance of these approaches under various conditions. In the absence of benchmarks for this research area, we introduce a set of our own based on the discretised variant of the DeepMind Control Suite (Tunyasuvunakool et al., 2020) used in prior value decomposition work (Seyde et al., 2022; Ireland & Montana, 2023). This benchmark contains transitions from agents trained to varying levels of performance across a range of diverse tasks, testing an agent's ability to learn complex behaviours from data of varying quality. In the spirit of advancing research in this area, we provide open access to these datasets as well as our full code base.

To the best of our knowledge, this investigation represents the first formative analysis of offline-RL in factorisable action spaces. We believe our work helps pave the way for developments in this important domain, whilst also contributing to the growing field of offline-RL more generally.

## 2 Preliminaries

### 2.1 Offline reinforcement learning

Following standard convention, we begin by defining a Markov Decision Process (MDP) with state space $\mathcal{S}$, action space $\mathcal{A}$, environment dynamics $T(s' \mid s, a)$, reward function $R(s, a)$ and discount factor $\gamma \in [0, 1]$ (Sutton & Barto, 2018). An agent interacts with this MDP by following a state-dependent policy $\pi(a \mid s)$, with the primary objective of discovering an optimal policy $\pi^*(a \mid s)$ that maximises the expected discounted sum of rewards, $\mathbb{E}_\pi \left[ \sum_{t=0}^{\infty} \gamma^t r(s_t, a_t) \right]$.

A popular approach for achieving this objective is through the use of Q-functions, $Q^\pi(s, a)$, which estimate the value of taking action $a$ in state $s$ and following policy $\pi$ thereafter. In discrete action spaces, optimal Q-values can be obtained by repeated application of the Bellman optimality equation:

$$Q^*(s, a) = r(s, a) + \gamma \mathbb{E}_{s' \sim T} \left[ \max_{a'} Q^*(s', a') \right] .$$

These Q-values can then be used to define an implicit policy such that:

$$\pi(s) = \arg\max_a Q(s, a) \ ,$$

*i.e.* the action that maximises the optimal Q-value at each state. Given the scale and complexity of real-world tasks, Q-functions are often parameterised (Mnih et al., 2013), with learnable parameters $\theta$ that are updated so as to minimise the following loss:

$$L(\theta) = \frac{1}{|B|} \sum_{(s,a,r,s') \in \mathcal{B}} (Q_\theta(s, a) - y)^2 \ , \tag{1}$$

where $y = r(s, a) + \gamma \max_{a'} Q_\theta(s', a')$ is referred to as the target value, and $B$ denotes a batch of data sampled uniformly at random from a replay buffer $\mathcal{B}$ of stored transitions from the agent's own interactions with the environment. To promote stability during training, when calculating Q-values in the target it is common to use a separate target network $Q_{\hat{\theta}}(s', a')$ (Mnih et al., 2013; Hessel et al., 2018) with parameters $\hat{\theta}$ updated towards $\theta$ either via a hard reset every specified number of steps, or gradually using Polyak-averaging.

In the offline setting, an agent is prohibited from interacting with the environment and must instead learn solely from a pre-existing dataset of interactions $\mathcal{B} = (s_b, a_b, r_b, s'_b)$, collected from some unknown behaviour policy (or policies) $\pi_\beta$ (Lange et al., 2012). This lack of interaction allows errors in Q-value estimates to compound and propagate during training, often resulting in a complete collapse of the learning process (Fujimoto et al., 2019b). Specifically, Q-values for out-of-distribution actions (*i.e.* those absent from the dataset) suffer from overestimation bias as a result of the maximisation carried out when determining target values. The outcome is specious Q-values estimates, and policies derived from those estimates consequently being highly sub-optimal. In order to compensate for this overestimation bias, Q-values must be regularised by staying "close" to the source data (Levine et al., 2020).

## 2.2 Decoupled Q-Network

By default, standard Q-learning approaches are based on atomic representations of action spaces (Sutton & Barto, 2018). This means that, in a factorisable action space, Q-values must be determined for every possible combination of sub-actions. This potentially renders such approaches highly ineffective due to combinatorial explosion in the number of atomic actions. Recalling that the action space can be thought of as a Cartesian product, then for each $\mathcal{A}_i$ we have that $|\mathcal{A}_i| = n_i$, and so the total number of atomic actions is $\prod_{i=1}^{N} n_i$. This quickly grows unwieldly as the number of sub-action spaces $N$ and/or number of actions within each sub-action space $n_i$ increase.

To address this issue, the Branching Dueling Q-Network (BDQ) proposed by Tavakoli et al. (2018) learns value estimates for each sub-action space independently and can be viewed as a single-agent analogue to independent Q-learning from multi-agent reinforcement learning (MARL) (Claus & Boutilier, 1998). Seyde et al. (2022) expand on this work with the introduction of the Decoupled Q-Network (DecQN), which computes value estimates in each sub-action space independently, but learns said estimates such that their mean estimates the Q-value for the combined (or global) action. Such an approach is highly reminiscent of the notion of value-decomposition used in cooperative MARL (Sunehag et al., 2017; Rashid et al., 2020b;a; Du et al., 2022), with sub-action spaces resembling individual agents.

In terms of specifics, DecQN introduces a utility function $U^i_{\theta_i}(s, a_i)$ for each sub-action space and redefines the Q-value to be:

$$Q_\theta(s, \mathbf{a}) = \frac{1}{N} \sum_{i=1}^{N} U^i_{\theta_i}(s, a_i) \ , \tag{2}$$

where $\mathbf{a} = (a_1, ..., a_N)$ is the global action, $\theta_i$ are the parameters for the $i$th utility function and $\theta = \{\theta_i\}_{i=1}^{N}$ are the global set of parameters. The loss in Equation (1) is updated to incorporate this utility function structure:

$$L(\theta) = \frac{1}{|B|} \sum_{(s,\mathbf{a},r,s') \in \mathcal{B}} (Q_\theta(s, \mathbf{a}) - y)^2 \ , \tag{3}$$

where

$$y = r(s, a) + \frac{\gamma}{N} \sum_{i=1}^{N} \max_{a_i'} U_{\theta_i}^i(s', a_i') \ .$$

As each utility function only needs to learn about actions within its own sub-action space, this reduces the total number of actions for which a value must be learnt to $\sum_{i=1}^{N} n_i$, thus preserving the functionality of established Q-learning algorithms. Whilst there are other valid decomposition methods, in this work we focus primarily on the decomposition proposed in DecQN. In Appendix D we provide a small ablation justifying our choice.

## 3 Related Work

### 3.1 Offline RL

Numerous approaches have been proposed to help mitigate Q-value overestimation bias in offline-RL. In BCQ (Fujimoto et al., 2019b), this is achieved by cloning a behaviour policy and using generated actions to form the basis of a policy which is then optimally perturbed by a separate network. BEAR (Kumar et al., 2019), BRAC (Wu et al., 2019) and Fisher-BRC (Kostrikov et al., 2021a) also make use of cloned behaviour policies, but instead use them to minimise divergence metrics between the learned and cloned policy. One-step RL (Brandfonbrener et al., 2021) explores the idea of combining fitted Q-evaluation and various policy improvement methods to learn policies without having to query actions outside the data. This is expanded upon in Implicit Q-learning (IQL) (Kostrikov et al., 2021b), which substitutes fitted Q-evaluation with expectile regression. TD3-BC (Fujimoto & Gu, 2021) adapts TD3 (Fujimoto et al., 2018) to the offline setting by directly incorporating behavioural cloning into policy updates, with TD3-BC-N/SAC-BC-N (Beeson & Montana, 2024) employing ensembles of Q-functions for uncertainty estimation to alleviate issues relating to overly restrictive constraints as well as computational burden present in other ensembles based approaches such as SAC-N & EDAC (An et al., 2021), MSG (Ghasemipour et al., 2022), PBRL (Bai et al., 2022) and RORL (Yang et al., 2022).

In the majority of cases the focus is on continuous action spaces, and whilst there have been adaptations and implementations in discrete action spaces (Fujimoto et al., 2019a; Gu et al., 2022), these tend to only consider a small number of (atomic) actions. This is also reflected in benchmark datasets such as D4RL (Fu et al., 2020) and RL Unplugged (Gulcehre et al., 2020). Our focus is on the relatively unexplored area of factorisable discrete action spaces.

### 3.2 Action decomposition

Reinforcement learning algorithms have been extensively studied in scenarios involving large, discrete action spaces. In order to overcome the challenges inherent in such scenarios, numerous approaches have been put forward based on action sub-sampling (Van de Wiele et al., 2020; Hubert et al., 2021), action embedding (Dulac-Arnold et al., 2015; Gu et al., 2022) and curriculum learning (Farquhar et al., 2020). However, such approaches are tailored to handle action spaces comprising numerous atomic actions, and do not inherently tackle the complexities nor utilise the structure posed by factorisable actions.

For factorisable action spaces various methods have been proposed, such as learning about sub-actions independently via value-based (Sharma et al., 2017; Tavakoli et al., 2018) or policy gradient methods (Tang & Agrawal, 2020; Seyde et al., 2021). Others have also framed the problem of action selection in factorisable action spaces as a sequence prediction problem, where the sequence consists of the individual sub-actions (Metz et al., 2017; Pierrot et al., 2021; Chebotar et al., 2023).

There exists a strong connection between factorisable action spaces and MARL, where the selection of a sub-action can be thought of as an individual agent choosing its action in a multi-agent setting. Value-decomposition has been shown to be an effective approach in MARL (Sunehag et al., 2017; Rashid et al., 2020b;a; Du et al., 2022), utilising the *centralised training with decentralised execution* paradigm (Kraemer & Banerjee, 2016), which allows agents to act independently but learn collectively. DecQN (Seyde et al., 2022)

and REValueD (Ireland & Montana, 2023) have subsequently shown such ideas can be used with factorised action spaces in single-agent reinforcement learning, demonstrating strong performance on a range of tasks that vary in complexity. Theoretical analysis of errors in Q-value estimates has also been conducted in efforts to stabilise training (Ireland & Montana, 2023; Thrun & Schwartz, 1993)

In this work, we focus on adapting DecQN to the offline setting by incorporating existing offline techniques. Whilst prior work has explored offline RL with value decomposition (Tang et al., 2022), this was limited to specific low-dimensional healthcare applications using only BCQ. Furthermore, accurately evaluating performance in such domains is notorious challenging (Gottesman et al., 2018). In contrast, we systematically study multiple offline methods using low and high-dimensional factorised action spaces across a suite of benchmark tasks.

## 4    Algorithms

In this Section we introduce several algorithms incorporating offline-RL methods into DecQN. We focus on methods that offer distinct takes on combatting overestimation bias, namely, policy constraints, conservative value estimation, implicit Q-learning and one-step RL. In each case, attention shifts from Q-values to utility values, with regularisation performed at the sub-action level.

### 4.1    DecQN-BCQ

Batch Constrained Q-learning (BCQ) (Fujimoto et al., 2019b;a) is a policy constraint approach to offline-RL. To compensate for overestimation bias in out-of-distribution actions, a cloned behaviour policy $\pi_\phi$ is used to restrict the actions available for target Q-values estimates, such that their probability under the behaviour policy meets a relative threshold $\tau$. This can be adapted and incorporated into DecQN by cloning separate behaviour policies $\pi_{\phi_i}^i$ for each sub-action dimension and restricting respective sub-actions available for corresponding target utility value-estimates.

The target value from Equation (3) becomes:

$$y = r(s, a) + \frac{\gamma}{N} \sum_{i=1}^{N} \max_{a_i' \, : \, \rho^i(a_i') \geq \tau} U_{\theta_i}^i(s', a_i') \, ,$$

where $\rho^i(a_i') = \pi_{\phi_i}^i(a_i' \mid s') / \max_{\hat{a}_i'} \pi_{\phi_i}^i(\hat{a}_i' \mid s')$ is the relative probability of sub-action $a_i'$ under policy $\pi_{\phi_i}^i$. Each cloned behaviour policy is trained via supervised learning with $\phi = \{\phi\}_{i=1}^{N}$. The full procedure can be found in Algorithm 1.

---

**Algorithm 1** DecQN-BCQ

---

**Require:** Threshold $\tau$, discounter factor $\gamma$, target network update rate $\mu$, number sub-action spaces $N$ and dataset $\mathcal{B}$.

    Initialise utility function parameters $\theta = \{\theta_i\}_{i=1}^{N}$, corresponding target parameters $\hat{\theta} = \theta$ and policy parameters $\phi = \{\phi_i\}_{i=1}^{N}$

    **for** $t = 0$ to $T$ **do**

        Sample minibatch of transitions $(s, \mathbf{a}, r, s')$ from $\mathcal{B}$

        $\phi \leftarrow \arg\min_\phi \frac{1}{N} \sum_{i=1}^{N} - \sum_{s, a_i} \log \pi_{\phi_i}^i(a_i \mid s)$

        $\theta \leftarrow \arg\min_\theta \sum_{s, \mathbf{a}, r, s'} (Q_\theta(s, \mathbf{a}) - y)^2$

        where:

          $Q_\theta(s, \mathbf{a}) = 1/N \sum_{i=1}^{N} U_{\theta_i}^i(s, a_i)$,

          $y = r + \gamma/N \sum_{i=1}^{N} \max_{a_i' \, : \, \rho^i(a_i') \geq \tau} U_{\hat{\theta}_i}^i(s', a_i')$,

          $\rho^i(a_i') = \pi_{\phi_i}^i(a_i' \mid s') / \max_{\hat{a}_i'} \pi_{\phi_i}^i(\hat{a}_i' \mid s')$

        $\hat{\theta} \leftarrow \mu\theta + (1 - \mu)\hat{\theta}$

    **end for**

---

## 4.2 DecQN-CQL

Conservative Q-learning (CQL) (Kumar et al., 2020) attempts to combat overestimation bias by targeting Q-values directly. The loss in Equation (1) is augmented with a term that "pushes-up" on Q-value estimates for actions present in the dataset and "pushes-down" for all others. This can be adapted and incorporated into DecQN by "pushing-up" on utility value estimates for sub-actions present in data and "pushing-down" for all others.

Using one particular variant of CQL this additional loss term under DecQN becomes:

$$L_{CQL}(\theta) = \frac{\alpha}{|B|} \sum_{s,\mathbf{a} \sim \mathcal{B}} \frac{1}{N} \sum_{i=1}^{N} \left[ \log \sum_{a_j \in \mathcal{A}_i} \exp(U_{\theta_i}^i(s, a_j)) - U_{\theta_i}^i(s, a_i) \right] ; \tag{4}$$

where $a_j$ denotes the $j$th sub-action within the $i$th sub-action space, and $\alpha$ is a hyperparameter that controls the overall level of conservatism. The full procedure can be found in Algorithm 2.

---

**Algorithm 2** DecQN-CQL

---

**Require:** Conservative coefficient $\alpha$, discounter factor $\gamma$, target network update rate $\mu$, number sub-action spaces $N$ and dataset $\mathcal{B}$.
    Initialise utility function parameters $\theta = \{\theta_i\}_{i=1}^N$ and corresponding target parameters $\hat{\theta} = \theta$
    **for** $t = 0$ to $T$ **do**
        Sample minibatch of transitions $(s, \mathbf{a}, r, s')$ from $\mathcal{B}$
        $\theta \leftarrow \arg\min_\theta \sum_{s,\mathbf{a},r,s'} (Q_\theta(s, \mathbf{a}) - y)^2 + \alpha \log \sum_j \exp Q_\theta(s, \mathbf{a_j}) - Q_\theta(s, \mathbf{a})$
        where:
          $Q_\theta(s, \mathbf{a}) = 1/N \sum_{i=1}^N U_{\theta_i}^i(s, a_i)$,
          $y = r + 1/N \sum_{i=1}^N \max_{a_i'} U_{\hat{\theta}_i}^i(s', a_i')$,
          $\log \sum_j \exp Q_\theta(s, \mathbf{a_j}) = 1/N \sum_{i=1}^N \log \sum_{a_j \in \mathcal{A}_i} \exp(U_{\theta_i}^i(s, a_j))$
        $\hat{\theta} \leftarrow \mu\theta + (1 - \mu)\hat{\theta}$
    **end for**

---

## 4.3 DecQN-IQL

Implicit Q-learning (IQL) (Kostrikov et al., 2021b) addresses the challenge of overestimation bias by learning a policy without having to query actions absent from data. A state and state-action value function are trained on the data and then used to extract a policy via advantage-weighted-behavioural-cloning.

The state value function $V_\psi(s)$ is trained via expectile regression, minimising the following loss:

$$L(\psi) = \frac{1}{|B|} \sum_{s,a \sim \mathcal{B}} [L_2^\tau(Q_\theta(s, a) - V_\psi(s))] ;$$

where if we denote $u = Q_\theta(s, a) - V_\psi(s)$ then $L_2^\tau(u) = |\tau - 1(u < 0)|u^2$ is the asymmetric least squares for the $\tau \in (0, 1)$ expectile.

The state-action value function $Q_\theta(s, a)$ is trained using the same loss as Equation (1), with the target value now $y = r(s, a) + \gamma V_\psi(s')$.

The policy follows that of discrete-action advantage-weighted-behavioural-cloning (Luo et al., 2023) such that:

$$\pi = \arg\max_a \left[ \frac{1}{\lambda} A(s, a) + \log \pi_\phi(a \mid s) \right] ;$$

where $A(s, a) = Q(s, a) - V(s)$ is the advantage function, $\pi_\phi(a \mid s)$ is a cloned behaviour policy trained via supervised learning and $\lambda$ is a hyperparameter controlling the balance between reinforcement learning and behavioural cloning.

This can be adapted and incorporated into DecQN by replacing $Q(s, a)$ with its decomposed form as per Equation (2) and adjusting the policy to reflect a sub-action structure, i.e:

$$\pi_i = \arg\max_{a_i} \left[ \frac{1}{\lambda} A(s, a_i) + \log \pi_{\phi_i}^i(a_i \mid s) \right] .$$

The full procedure can be found in Algorithm 3.

---

**Algorithm 3** DecQN-IQL
___
**Require:** Expectile $\tau$, discounter factor $\gamma$, target network update rate $\mu$, number sub-action spaces $N$ and dataset $\mathcal{B}$.
  Initialise utility function parameters $\theta = \{\theta_i\}_{i=1}^N$ and corresponding target parameters $\hat{\theta} = \theta$. Initialise state value function parameters $\psi$ and policy parameters $\phi = \{\phi_i\}_{i=1}^N$
  **for** $t = 0$ to $T$ **do**
    Sample minibatch of transitions $(s, \mathbf{a}, r, s')$ from $\mathcal{B}$
    $\phi \leftarrow \arg\min_\phi \frac{1}{N} \sum_{i=1}^N - \sum_{s,a_i} \log \pi_{\phi_i}^i(a_i \mid s)$
    $\theta \leftarrow \arg\min_\theta \frac{1}{N} \sum_{s,\mathbf{a},r,s'} (Q_\theta(s, \mathbf{a}) - y)^2$
    $\psi \leftarrow \arg\min_\psi \frac{1}{N} \sum_{s,\mathbf{a}} [L_2^\tau(Q_{\hat{\theta}}(s, \mathbf{a}) - V_\psi(s))]$
    where:
      $Q_\theta(s, \mathbf{a}) = 1/N \sum_{i=1}^N U_{\theta_i}^i(s, a_i),$
      $Q_{\hat{\theta}}(s, \mathbf{a}) = 1/N \sum_{i=1}^N U_{\hat{\theta}_i}^i(s, a_i),$
      $y = r + V_\psi(s')$
    $\hat{\theta} \leftarrow \mu\theta + (1 - \mu)\hat{\theta}$
  **end for**
___

## 4.4 DecQN-OneStep

We can derive an alternative approach to IQL which removes the requirement for a separate state value function altogether. Noting that $V(s) = \sum_a \pi(a \mid s) Q(s, a)$, we can instead use the cloned behaviour policy $\pi_\phi(a' \mid s')$ and state-action value function $Q_\theta(s', a')$ to calculate the state value function $V(s')$ instead. This can be adapted and incorporated into DecQN be replacing $Q(s, a)$ with its decomposed form as per Equation (2) and adjusting the policy to reflect a sub-action structure. We denote this approach DecQN-OneStep as it mirrors one-step RL approaches that train state value functions using fitted Q-evaluation (Brandfonbrener et al., 2021). The full procedure can be found in Algorithm 4.

---

**Algorithm 4** DecQN-OneStep
___
**Require:** Discounter factor $\gamma$, target network update rate $\mu$, number sub-action spaces $N$ and dataset $\mathcal{B}$.
  Initialise utility function parameters $\theta = \{\theta_i\}_{i=1}^N$ and corresponding target parameters $\hat{\theta} = \theta$. Initialise policy parameters $\phi = \{\phi_i\}_{i=1}^N$
  **for** $t = 0$ to $T$ **do**
    Sample minibatch of transitions $(s, \mathbf{a}, r, s')$ from $\mathcal{B}$
    $\phi \leftarrow \arg\min_\phi \frac{1}{N} \sum_{i=1}^N - \sum_{s,a_i} \log \pi_{\phi_i}^i(a_i \mid s)$
    $\theta \leftarrow \arg\min_\theta \frac{1}{N} \sum_{s,\mathbf{a},r,s'} (Q_\theta(s, \mathbf{a}) - y)^2$
    where:
      $Q_\theta(s, \mathbf{a}) = 1/N \sum_{i=1}^N U_{\theta_i}^i(s, a_i),$
      $y = r + 1/N \sum_{i=1}^N \sum_{a_i} \pi_{\phi_i}^i(a_i \mid s) U_{\hat{\theta}_i}^i(s', a_i')$
    $\hat{\theta} \leftarrow \mu\theta + (1 - \mu)\hat{\theta}$
  **end for**
___

## 5 Environments and datasets

Although there are a number of established environments/tasks for continuous and small-scale discrete action spaces, there is an absence of similar environments/tasks specifically designed for factorisable action spaces. As such, there is also an absence of benchmark datasets analogous to those found in D4RL (Fu et al., 2020) and RL Unplugged (Gulcehre et al., 2020). In light of this, we introduce our own set of benchmarks based on a discretised variant of the DeepMind control suite, as previously adopted by Seyde et al. (2022); Ireland & Montana (2023). This suite contains a variety of environments and tasks that range in size and complexity, which although originally designed for continuous control (which is not our focus), can easily be repurposed for a discrete factorisable setting by discretising individual sub-action spaces. This discretisation process involves selecting a subset of actions from the original continuous space which then become the discrete actions available to the agent. For example, a continuous action in the range $[-1, 1]$ can be discretised into three discrete actions corresponding to the subset $\{-1, 0, 1\}$. We emphasise this discretisation procedure happens prior to data collection.

For the datasets themselves, we follow a similar procedure to D4RL. Using DecQN/REValueD, we train agents to "expert" and "medium" levels of performance and then collect 1M transitions from the resulting policies. Here, we define "expert" to be the peak performance achieved by DecQN/REValueD and "medium" to be approximately 1/3rd the performance of the "expert". We create a third dataset "medium-expert" by combining transitions from these two sources and a fourth "random-medium-expert" containing 200k transitions constituting 45% random and medium transitions and 10% expert. Each of these datasets presents a specific challenge to agents, namely the ability to learn from optimal or sub-optimal data ("expert" and "medium", respectively) as well as data that contains a mixture ("medium-expert" and "random-medium-expert"). More details on this training and data collection procedure are provided in Appendix A.

## 6 Experimental evaluation

### 6.1 Implementation

We train agents using DecQN, DecQN-BCQ, DecQN-CQL, DecQN-IQL and DecQN-OneStep on our benchmark datasets and evaluate their performance in the simulated environment. We also train and evaluate agents using a factorised equivalent of behavioural cloning to provide a supervised learning baseline.

Utility functions are parameterised by neural networks, comprising a 2-layer MLP with ReLU activation functions and 512 nodes, taking in a normalised state as input and outputting utility values for each sub-action space. We use the same architecture for policies, with the output layer a softmax across actions within each sub-actions-space. State value functions mirror this architecture except in the final layer which outputs a single value. We train networks via stochastic gradient descent using the Adam optimiser (Kingma & Ba, 2014) with learning rate $3e^{-4}$ and a batch size of 256. For state and state-action value functions we use the Huber loss as opposed to MSE loss. We set the discount factor $\gamma = 0.99$ and the target network update rate $\mu = 0.005$. We utilise a dual-critic approach, taking the mean across two utility estimates for target Q-values. All agents are trained for 1M gradient updates.

The only hyperparameters we tune are the threshold $\tau$ in BCQ, conservative coefficent $\alpha$ in CQL, expectile $\tau$ and balance coefficient $\lambda$ in IQL/OneStep. We allow these to vary across environment/task, but to better reflect real-world scenarios where the quality of data may be unknown, we forbid variation within environments/tasks. Values for each environment/task dataset can be found in Table 3 in the Appendix.

Performance is measured in terms of normalised score, where:

$$score_{norm} = 100 \times \frac{score - score_{random}}{score_{expert} - score_{random}} \; ;$$

with 0 representing a random policy and 100 the "expert" policy from the fully trained agent. We repeat experiments across five random seeds, and evaluate each agent's final policy 10 times, reporting results as mean normalised scores $\pm$ one standard error across seeds. For each set of experiments we provide visual summaries with tabulated results available in Appendix C for completeness.

## 6.2 Case study: Atomic-CQL vs DecQN-CQL

Before evaluating and comparing agents on the full benchmark, we conduct a case study using CQL which directly compares performance and computation using a standard atomic representation of actions, which we denote Atomic-CQL, and a factorised and decomposed approach as proposed in DecQN-CQL.

Following the procedure outlined in Section 5 we construct a "medium-expert" dataset of 100k transitions for the "cheetah-run" task for bin sizes $n_i \in \{3, 4, 5, 6\}$. We train agents using Atomic-CQL and DecQN-CQL for 500k gradient steps and compare both the performance of resulting policies and overall computation time. For both Atomic-CQL and DecQN-CQL we set $\alpha = 0.5$ for $n = 3$ and $\alpha = 2$ for $n \in \{4, 5, 6\}$. We summarise results in Figure 1.

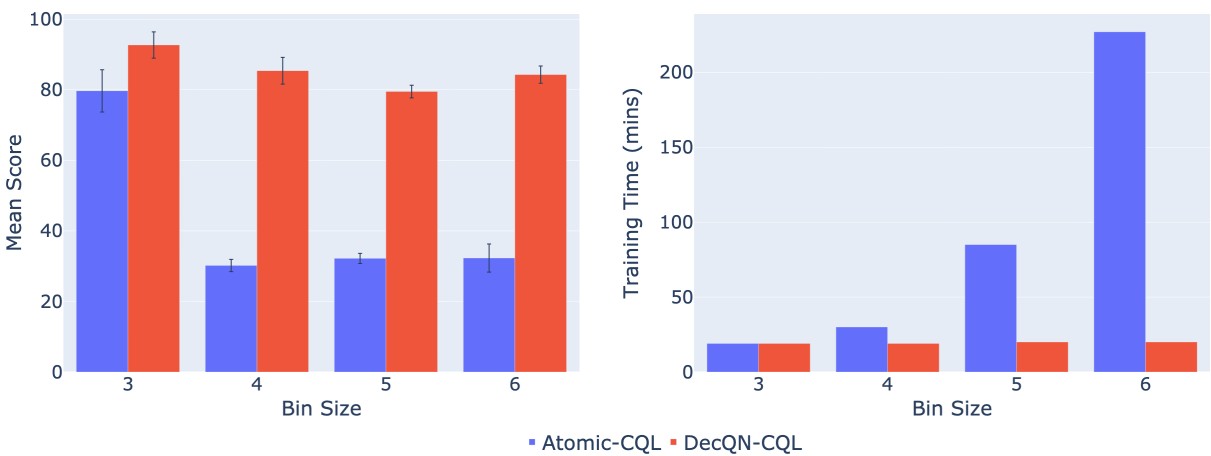

Figure 1: Comparisons of performance (left) and computation time (right) for Atomic-CQL and DecQN-CQL on the "cheetah-run-medium-expert" dataset for varying bin-sizes. Performance is measured in terms of mean normalised score $\pm$ one standard error, where 0 and 100 represent random and expert policies, respectively. As bin-size increases, Atomic-CQL suffers notable drops in performance and increases in computation time, whereas DecQN-CQL is relatively resilient in both areas.

We see that as the number of sub-actions $n_i$ increases, Atomic-CQL exhibits a notable decline in performance, whereas DecQN-CQL performance declines only marginally. We also see a dramatic increase in computation time for Atomic-CQL whereas DecQN-CQL remains roughly constant. For Atomic-CQL, these outcomes are symptomatic of the combinatorial explosion in the number of actions requiring value-estimation and the associated number of out-of-distribution global actions. These issues are less prevalent in DecQN-CQL due to its factorised and decomposed formulation.

To provide further insights, we also examine the evolution of Q-value errors during training. Every 5k gradient updates we obtain a MC estimate of true Q-values using discounted rewards from environmental rollouts. We then compare these MC estimates with Q-values predicted by both Atomic-CQL and DecQN-CQL networks for the respective actions taken. To make better use of rollouts (which can be expensive), we calculate MC estimates and Atomic-/DecQN-CQL Q-values for the first 500 states in the trajectory, as using a discount factor of $\gamma = 0.99$ discounts rewards by over 99% for time-steps beyond 500 (and all tasks considered have trajectory length 1000). In total we perform 10 rollouts, giving 5000 estimates of the error between true and Atomic-CQL/DecQN-CQL Q-values. In Figure 2 we plot the mean absolute error over the course of training, with the solid line representing the mean across five random seeds and shaded area the standard error. For all values of $n_i$ we observe the mean absolute error is less for DecQN-CQL than Atomic-CQL, particularly for $n_i > 3$, aligning with each algorithm's respective performance in Figure 1.

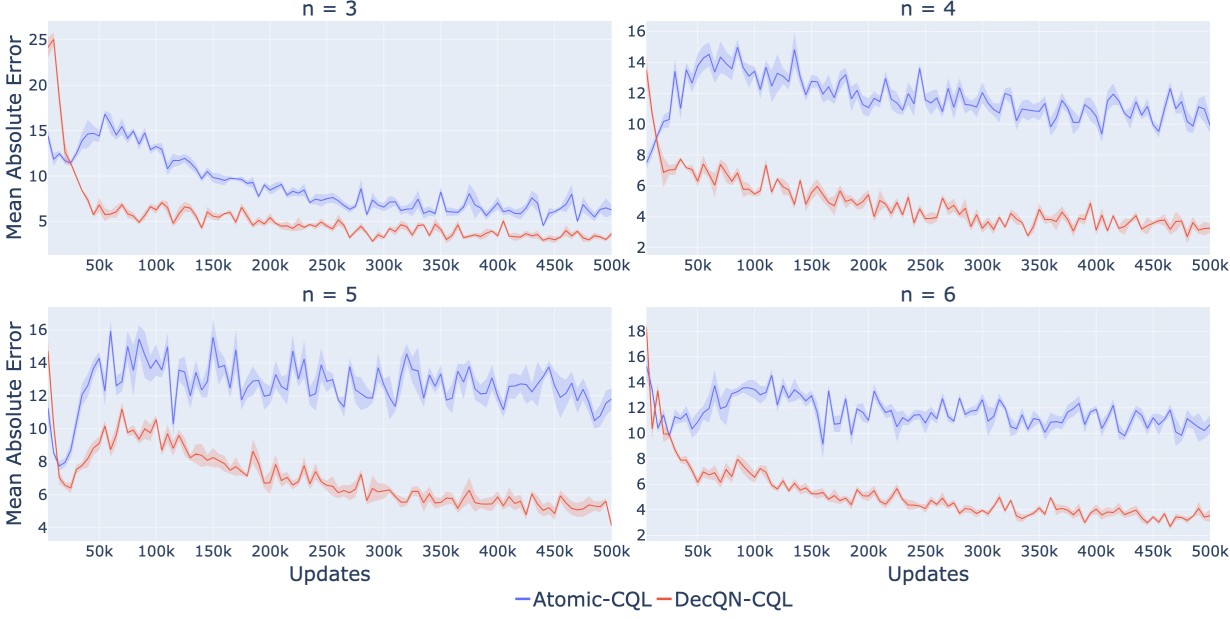

Figure 2: Comparison of estimated errors in Q-values for "cheetah-run-medium-expert" dataset for varying bin sizes. Errors are lower for DecQN-CQL for all bin-sizes, most notably for $n_i > 3$, mirroring the deviation in performance levels between the two approaches.

## 6.3 Benchmark comparison

We train and evaluate agents across our benchmark suite setting $n_i = 3$. This necessitates the use of value-decomposition for all but the most simple tasks, as highlighted in Table 1 where we summarise each environment's state and action space. Results are summarised in Figure 3.

Table 1: Environment details for DeepMind Control Suite. $|S|$ represents the size of the state space and $N$ the number of sub-action spaces. $\prod_i n_i$ is the total number of actions under atomic representation and $\sum_i n_i$ under factorised representation when $n_i = 3$.

| Environment | $|S|$ | $N$ | $\prod_i n_i$ | $\sum_i n_i$ |
|---|---|---|---|---|
| Finger Spin | 9 | 2 | 9 | 6 |
| Fish Swim | 24 | 5 | 243 | 15 |
| Cheetah Run | 17 | 6 | 729 | 18 |
| Quadruped Walk | 78 | 12 | $\approx 530k$ | 36 |
| Humanoid Stand | 67 | 21 | $\approx 10^{10}$ | 63 |
| Dog Trot | 223 | 38 | $\approx 10^{18}$ | 114 |

In general we see that all offline-RL methods outperform behavioural cloning across all environments/tasks and datasets, with the exception of DecQN-BCQ for "random-medium-expert" datasets which performs quite poorly. DecQN without offline adjustments leads to highly sub-optimal policies, in the vast majority of cases failing to learn a policy that improves over random behaviour (a direct consequence of aforementioned issues relating to overestimation bias). In terms of offline methods specifically, in general DecQN-CQL has a slight edge over others for lower dimensional tasks such as "finger-spin", "fish-swim" and "cheetah-run", whilst DecQN-IQL/OneStep have the edge for higher-dimensional tasks such as "humanoid-stand" and "dog-trot". For "medium-expert" datasets we see in most cases all methods are able to learn expert or near-expert level

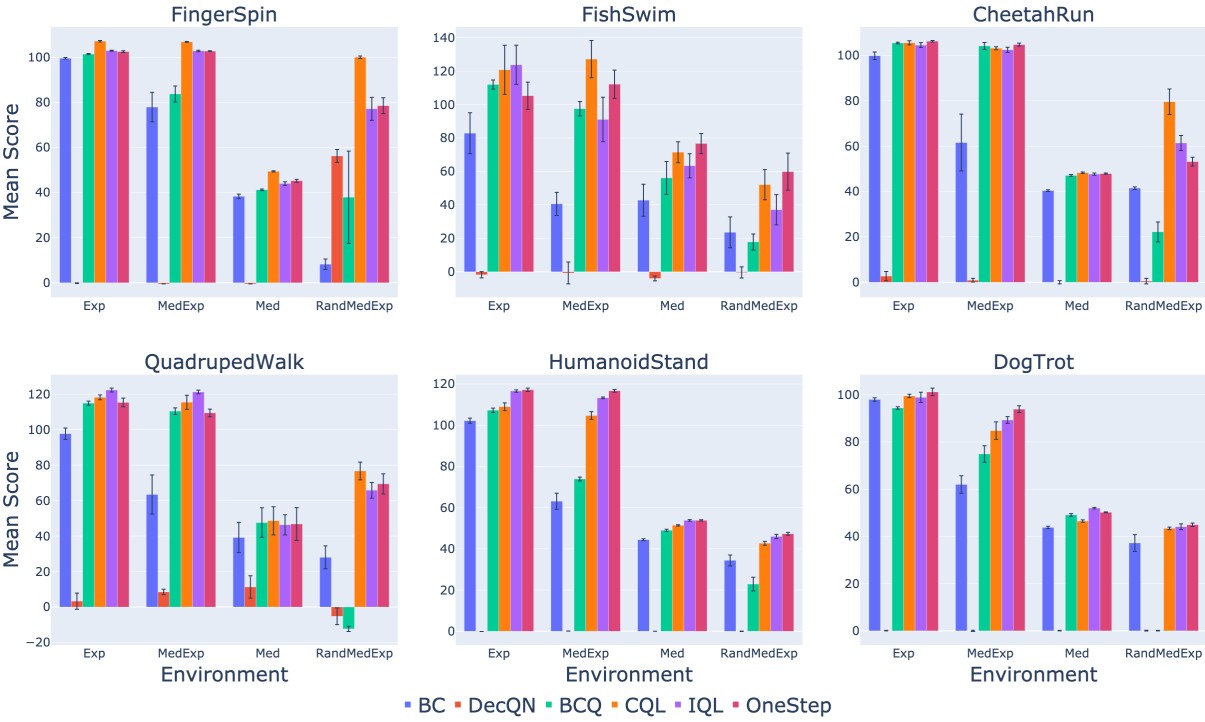

Figure 3: Performance comparison across benchmark for $n_i = 3$. For presentation purposes the prefix "DecQN-" has been omitted for each offline method. Figures are mean normalised score $\pm$ one standard error, where 0 and 100 represent random and expert policies, respectively. In general, all offline-RL methods improve over behavioural cloning, with the exception of DecQN-BCQ for random-medium-expert datasets. DecQN without any offline modification performs poorly across all environments/tasks.

policies. Extracting optimal behaviour from "random-medium-expert" datasets proves significantly more challenging, likely a result of these datasets being both highly variable and constituting relatively few expert trajectories.

## 6.4  Larger bin sizes

In this sub-section we investigate the impact of increasing the number of sub-actions within each sub-action dimension. This helps provide insights into the ability of our chosen offline methods to scale to larger and larger action spaces. We focus in particular on the dog-trot environment since this is by far the largest in terms of actions. We collect datasets following the same procedure outlined in Section 5 for bin sizes $n_i \in \{10, 30, 50, 75, 100\}$. We summarise results in Figure 4.

In general, we see that our chosen offline methods are robust to increases in bin size, continuing to outperform behavioural cloning (with the same exception for DecQN-BCQ on "random-medium-expert") and extract near-expert policies from "medium-expert" datasets, with DecQN-IQL/-OneStep maintaining their edge over DecQN-CQL. For "random-medium-expert" datasets we start to notice a decline in performance as we approach the upper end of our bin size range, most noticeably when $n = 100$. This is likely a consequence of higher bin sizes exacerbating the difficulties in obtaining good policies from highly variable and largely sub-optimal data.

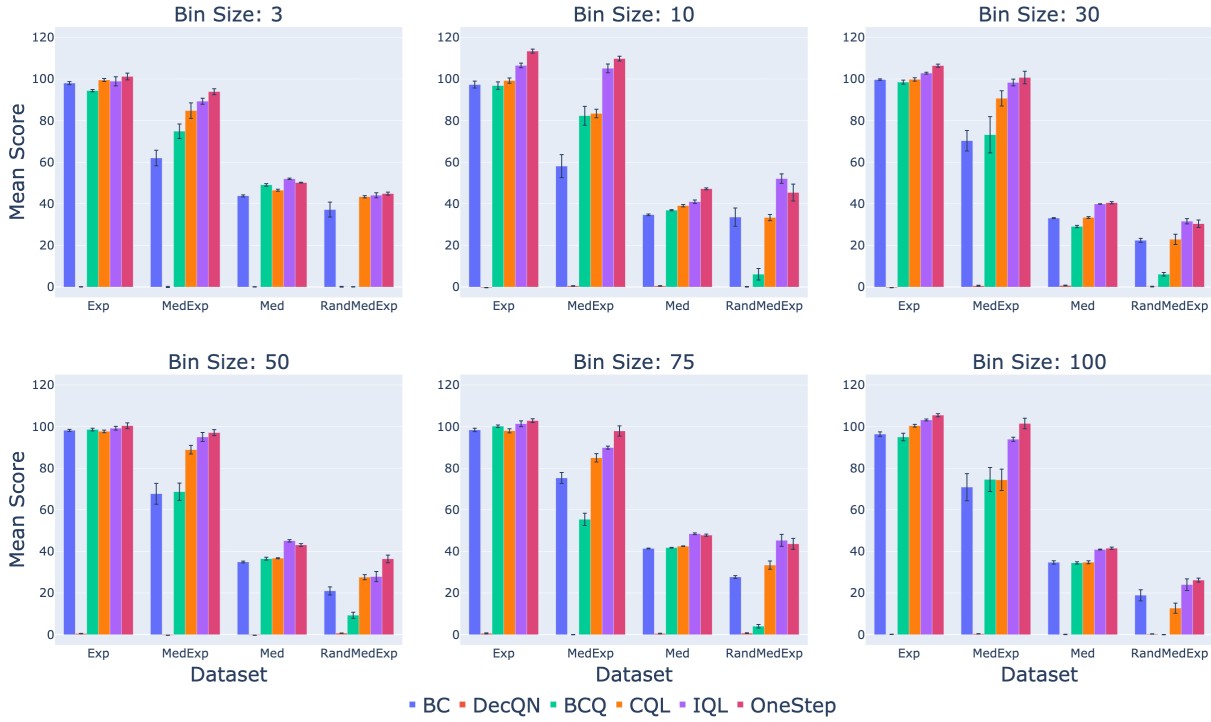

Figure 4: Performance comparison for dog-trot for $n_i \in \{3, 10, 30, 50, 75, 100\}$. For presentation purposes the prefix "DecQN-" has been omitted for each offline method. Figures are mean normalised score $\pm$ one standard error, where 0 and 100 represent random and expert policies, respectively. Each approach is reasonably resilient to larger bin sizes, although for "random-medium-expert" datasets extracting a good policy appears to become more challenging as $n$ gets very large.

# 7 Discussion and conclusion

In this work, we have explored the realm of offline reinforcement in factorisable action spaces. Through empirical evaluation, we have shown how a factorised and decomposed approach offers numerous benefits over standard/atomic approaches. Using a bespoke dataset we have conducted an extensive empirical evaluation of several offline-RL approaches adapted to this factorised and decomposed framework, providing insights into each approach's ability to learn tasks of varying complexity from datasets of differing size and quality.

In general, our empirical evaluation demonstrates our chosen offline methods adapt well to the factorised setting when combined with value-decomposition in the form of DecQN. With one exception, all approaches are consistently able to outperform behavioural cloning regardless of data quality, and where datasets contain sufficient levels of high-quality trajectories (*i.e.* "expert" and "medium-expert"), obtain expert/near-expert policies, even as the number of actions increase. There is however notable room for improvement for datasets with a scarcity of high-quality trajectories (*i.e.* "medium" and "random-medium-expert").

Our initial investigation opens up numerous other possibilities for future research. One of these is the development of techniques for automatically tuning hyperparameters during training, which at present are not environment/task agnostic. In addition, as with their continuous counterparts, performance can be enhanced by allowing hyperparameters to vary for each dataset (see Appendix B.1 for examples). Off-policy evaluation could also prove beneficial here (Rebello et al., 2023), providing assurances on the quality of a policy prior to deployment.

For DecQN-BCQ/-IQL/-OneStep, alternative approaches to modelling the behaviour policy $\pi_\phi$ may help improve performance for more challenging datasets. In our particular implementation we use an MLP, but alternative architectures such as LSTMs may better capture underlying environmental complexity (Scheller et al., 2020). Incorporating other methods outlined in Section 3 is another possibility. For example, the use of ensembles for capturing uncertainty in value estimates has been shown to perform well in combination with behavioural cloning in continuous action spaces (Beeson & Montana, 2024), and is a relatively straightforward extension to the approaches we consider here.

Whilst DecQN offers a simple, effective and computationally efficient foundation for offline-RL in factorisable action spaces, we note there are some inherent assumptions and limitations to the value decomposition approach that warrant further investigation. In particular, the efficacy of DecQN relies on individual sub-action optimisation leading to globally optimal joint policies. However, for tasks with sparse rewards or complex sub-action dependencies, individually learned sub-policies may fail to properly compose into a coherent overall policy. For example, in assembly tasks, separately learned pick, place, and connect skills could lead to conflicting behaviors when combined. Additional research into modeling sub-action interactions during decomposition could help overcome this limitation.

Owing to practical considerations, the discretisation procedure in this work is relatively simplistic, splitting actions into equally sized and equally spaced bins. Future work could investigate more nuanced aspects relating to action spaces, such as variable bin sizes, non-even spacing between actions, clustering of actions and masked actions. Furthermore, the creation of bespoke environments would be particularly beneficial, as this not only removes the need to discretise continuous-action environments, but provides more realistic scenarios to evaluate against.

We hope our work underscores the unique setting and challenges of conducting offline-RL in factorisable action spaces and paves the way for future research by providing an accessible and solid foundation from which to build upon.

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

# A   Data collection procedure

To collect the datasets we followed the training procedures laid out by (Seyde et al., 2022; Ireland & Montana, 2023) to train the Decoupled Q-networks. To expedite the data collection process during training we used a distributed setup using multiple workers to collect data in parallel (Horgan et al., 2018). We parameterised the utility functions using a (shared) single ResNet layer followed by layer norm, followed by a linear head for each of the sub-action spaces which predicts sub-action utility values. Full details regarding network architecture and hyperparameters can be found in Table 2.

Once we have trained the DecQNs we create the benchmark datasets by collecting data using a greedy policy derived from the learned utility values. Each dataset contains 1M transitions – as each episode is truncated at 1,000 time steps in the DM control suite, this corresponds to collecting 1,000 episodes. For the expert policy we trained the DecQNs until their test performance corresponded to the performance given in (Seyde et al., 2022; Ireland & Montana, 2023). For the medium policy we aimed for a test performance of approximately 1/3rd of the reported expert score.

We largely employ the same hyperparameters as the original DecQN study, as detailed in Table 3. Exceptions include the decay of the exploration parameter ($\epsilon$) to a minimum value instead of keeping it constant, and the use of Polyak-averaging for updating the target network parameters, as opposed to a hard reset after every specified number of updates. Finally, we sample from the replay buffer uniformly at random, as opposed to using a priority. We maintain the same hyperparameters across all our experiments.

For $n = 3$ we use DecQN to train networks and collect datasets. For $n > 3$ we use REValueD to train networks and collect datasets due to better scaling to larger bin sizes (Ireland & Montana, 2023).

Table 2: Hyperparameters used in DecQN and REValueD training.

| Parameters | Value |
|---|---|
| Optimizer | Adam |
| Learning rate | $1 \times 10^{-4}$ |
| Replay size | $5 \times 10^5$ |
| n-step returns | 3 |
| Discount, $\gamma$ | 0.99 |
| Batch size | 256 |
| Hidden size | 512 |
| Gradient clipping | 40 |
| Target network update parameter, $c$ | 0.005 |
| Imp. sampling exponent | 0.2 |
| Priority exponent | 0.6 |
| Minimum exploration, $\epsilon$ | 0.05 |
| $\epsilon$ decay rate | 0.99995 |
| Regularisation loss coefficient $\beta$ | 0.5 |
| Ensemble size K | 10 |

# B   Hyperparameters

Following on from Section 6, Table 3 provides hyperparameters for all environments/tasks. For DecQN-BCQ we searched over $\tau = \{0.05, 0.1, 0.25, 0.5\}$. For DecQN-CQL we searched over $\alpha = \{0.25, 0.5, 1, 2\}$. For DecQN-IQL we searched over $\tau = \{0.5, 0.6, 0.7, 0.8\}$, $\lambda = \{1, 2, 5, 10\}$. For DecQN-OneStep we searched over $\lambda = \{1, 2, 5, 10\}$.

Table 3: Hyperparameters for experiments in Section 6

| Environment/task | Bin Size ($n$) | BCQ $\tau$ | CQL $\alpha$ | IQL $\beta$, $\lambda$ | OneStep $\lambda$ |
|---|---|---|---|---|---|
| FingerSpin | 3 | 0.25 | 0.25 | 0.5, 1 | 1 |
| FishSwim | 3 | 0.25 | 0.25 | 0.5, 1 | 1 |
| CheetahRun | 3 | 0.25 | 0.25 | 0.5, 1 | 1 |
| QuadrupedWalk | 3 | 0.25 | 0.25 | 0.5, 2 | 1 |
| HumanoidStand | 3 | 0.25 | 0.25 | 0.5, 2 | 2 |
| DogTrot | 3 | 0.25 | 1 | 0.5, 5 | 5 |
| DogTrot | 10 | 0.25 | 0.25 | 0.5, 5 | 2 |
| DogTrot | 30 | 0.25 | 1 | 0.5, 2 | 2 |
| DogTrot | 50 | 0.5 | 0.5 | 0.5, 2 | 2 |
| DogTrot | 75 | 0.5 | 0.5 | 0.5, 2 | 2 |
| DogTrot | 100 | 0.5 | 2 | 0.5, 5 | 5 |

Table 4: Individual performance allowing hyperparameters to vary within environment/taskc; "dog-trot', $n = 3$. Figures are mean normalised scores, with 0 and 100 representing random and expert policies, respectively. Highest score highlighted in bold

| Environment -dataset | | | | |
|---|---|---|---|---|
| DogTrot (BCQ) | $\tau = 0.025$ | $\tau = 0.05$ | $\tau = 0.1$ | $\tau = 0.25$ |
| -expert | 57.8 | 85 | 93.6 | **94.4** |
| -medium-expert | 3.9 | 10.1 | 42.7 | **74.9** |
| -medium | 39.8 | 38.8 | 34.2 | **49.1** |
| -random-medium-expert | 5 | 5.6 | **9** | 0.1 |
| DogTrot (CQL) | $\alpha = 0.25$ | $\alpha = 0.5$ | $\alpha = 1$ | $\alpha = 2$ |
| -expert | 90.8 | 95.7 | 99.5 | **100.2** |
| -medium-expert | 76.6 | 81.7 | **84.8** | 75.1 |
| -medium | **50.6** | 48.3 | 46.5 | 45.2 |
| -random-medium-expert | 41.2 | 40.8 | **43.4** | 38.6 |
| DogTrot (IQL $\tau = 0.5$) | $\lambda = 1$ | $\lambda = 2$ | $\lambda = 5$ | $\lambda = 10$ |
| -expert | 37.9 | 82.5 | 98.9 | **99.5** |
| -medium-expert | 33 | 64 | 89.3 | **98.6** |
| -medium | **58.8** | 56.5 | 52 | 47.3 |
| -random-medium-expert | 10.6 | 28.6 | 44.1 | **44.7** |
| DogTrot (OneStep) | $\lambda = 1$ | $\lambda = 2$ | $\lambda = 5$ | $\lambda = 10$ |
| -expert | 53.3 | 91.7 | 101.2 | **102** |
| -medium-expert | 44.5 | 79.4 | 93.9 | **96.6** |
| -medium | **59.3** | 57.5 | 50.2 | 48 |
| -random-medium-expert | 23.8 | 43.9 | 44.9 | **45.1** |

### B.1 Allowing hyperparameter variation within environment/task

As per Section 7, in Table 4 we provide examples of performance improvement after permitting hyperparameter variation within the same environment/task. In general, we see lower quality datasets benefit from smaller hyperparameters (*i.e.* those that weight more towards RL and less towards BC) and higher quality datasets benefit from larger hyperparameters (*i.e.* those the weight more towards BC and less towards RL). This mirrors findings from previous papers outlined in Section 3.

## C   Tabulated results

Tabulated results for Figure 1 are presented in Table 5. Tabulated results for Figure 3 are presented in Table 6. Tabulated results for Figure 4 are presented in Table 7.

Table 5: Atomic-CQL vs DecQN-CQL - performance and computation comparison. Performance figures are mean normalised scores $\pm$ one standard error, with 0 and 100 representing random and expert policies, respectively. Computation figures are training time and GPU usage. Actions figures are total number of actions requiring value estimation based on atomic/factorised representation.

| Method | $n_i$ | Actions | Score | Training time (mins) | GPU usage (MB) |
|--------|-------|---------|-------|----------------------|----------------|
| Atomic-CQL | 3 | 729 | $79.7 \pm 6.0$ | 19 | 266 |
| Atomic-CQL | 4 | 4096 | $30.2 \pm 1.7$ | 30 | 412 |
| Atomic-CQL | 5 | 15625 | $32.2 \pm 1.4$ | 85 | 958 |
| Atomic-CQL | 6 | 46656 | $32.3 \pm 4.0$ | 227 | 2388 |
| DecQN-CQL | 3 | 18 | $92.7 \pm 3.7$ | 19 | 244 |
| DecQN-CQL | 4 | 24 | $85.4 \pm 3.8$ | 19 | 246 |
| DecQN-CQL | 5 | 30 | $79.5 \pm 1.8$ | 20 | 246 |
| DecQN-CQL | 6 | 36 | $84.3 \pm 2.5$ | 20 | 246 |

Table 6: Individual performance comparison $n = 3$. Figures are mean normalised scores $\pm$ one standard error, with 0 and 100 representing random and expert policies, respectively. At the bottom of the table we also provide totals, split by data quality and the entire benchmark. We see that DecQN-BCQ is the least performant of the offline methods and that DecQN-CQL/IQL/OneStep perform similarly overall and on expert, medium-exert and and medium datasets and DecQN-CQL the best on random-medium-expert datasets.

| Environment -dataset | BC | DecQN | DecQN-BCQ | DecQN-CQL | DecQN-IQL | DecQN-OneStep |
|---|---|---|---|---|---|---|
| **FingerSpin** | | | | | | |
| -expert | $99.5 \pm 0.4$ | $-0.2 \pm 0.1$ | $101.4 \pm 0.3$ | $107.1 \pm 0.3$ | $102.9 \pm 0.2$ | $102.5 \pm 0.4$ |
| -medium-expert | $77.9 \pm 6.5$ | $-0.5 \pm 0$ | $83.7 \pm 3.5$ | $106.8 \pm 0.2$ | $102.8 \pm 0.3$ | $102.7 \pm 0.2$ |
| -medium | $38.3 \pm 1$ | $-0.5 \pm 0$ | $41.2 \pm 0.4$ | $49.4 \pm 0.3$ | $44 \pm 0.8$ | $45.2 \pm 0.6$ |
| -random-medium-expert | $8.2 \pm 2.3$ | $56.2 \pm 2.9$ | $37.9 \pm 20.4$ | $100 \pm 0.5$ | $77.1 \pm 5.1$ | $78.5 \pm 3.5$ |
| **FishSwim** | | | | | | |
| -expert | $82.9 \pm 12.2$ | $-1.8 \pm 1.9$ | $112 \pm 2.7$ | $120.8 \pm 14.7$ | $123.8 \pm 11.7$ | $105.3 \pm 8.2$ |
| -medium-expert | $40.6 \pm 6.9$ | $0.7 \pm 6.5$ | $97.5 \pm 4.3$ | $127.2 \pm 11.1$ | $91.1 \pm 13.3$ | $112.2 \pm 8.5$ |
| -medium | $42.8 \pm 9.6$ | $-4 \pm 1.5$ | $56.1 \pm 9.8$ | $71.5 \pm 6.3$ | $63.4 \pm 7.2$ | $76.7 \pm 6$ |
| -random-medium-expert | $23.6 \pm 9.2$ | $-0.4 \pm 3.3$ | $17.8 \pm 4.8$ | $52.1 \pm 9$ | $37.1 \pm 9$ | $59.9 \pm 11$ |
| **CheetahRun** | | | | | | |
| -expert | $99.9 \pm 1.7$ | $2.7 \pm 2.1$ | $105.5 \pm 0.4$ | $105.6 \pm 0.9$ | $104.6 \pm 1.1$ | $106.3 \pm 0.3$ |
| -medium-expert | $61.6 \pm 12.5$ | $0.9 \pm 0.8$ | $104.2 \pm 1.6$ | $103.2 \pm 0.7$ | $102.5 \pm 1.2$ | $104.8 \pm 0.7$ |
| -medium | $40.4 \pm 0.4$ | $0 \pm 0.7$ | $47.1 \pm 0.4$ | $48.3 \pm 0.3$ | $47.7 \pm 0.5$ | $47.9 \pm 0.3$ |
| -random-medium-expert | $41.5 \pm 0.6$ | $0.5 \pm 1.2$ | $22.2 \pm 4.4$ | $79.6 \pm 5.6$ | $61.4 \pm 3.3$ | $53.2 \pm 1.9$ |
| **QuadrupedWalk** | | | | | | |
| -expert | $97.7 \pm 3.2$ | $3.3 \pm 4.6$ | $114.9 \pm 1.2$ | $118.2 \pm 1.4$ | $122.3 \pm 1.1$ | $115.3 \pm 2.4$ |
| -medium-expert | $63.4 \pm 11$ | $8.5 \pm 1.5$ | $110.4 \pm 1.9$ | $115.4 \pm 4$ | $121.2 \pm 1$ | $109.4 \pm 2.1$ |
| -medium | $39.2 \pm 8.5$ | $11.3 \pm 6.3$ | $47.6 \pm 8.4$ | $48.6 \pm 7.9$ | $46.3 \pm 5.7$ | $46.8 \pm 9.2$ |
| -random-medium-expert | $28 \pm 6.4$ | $-5.2 \pm 4.7$ | $-12.3 \pm 1.5$ | $76.7 \pm 5$ | $65.8 \pm 4.4$ | $69.4 \pm 5.7$ |
| **HumanoidStand** | | | | | | |
| -expert | $102.2 \pm 1.3$ | $-0.1 \pm 0$ | $107.3 \pm 1$ | $109 \pm 1.9$ | $116.6 \pm 0.6$ | $117.2 \pm 0.8$ |
| -medium-expert | $63.1 \pm 3.9$ | $0.1 \pm 0$ | $73.9 \pm 0.9$ | $104.7 \pm 1.9$ | $113.3 \pm 0.4$ | $116.7 \pm 0.7$ |
| -medium | $44.4 \pm 0.5$ | $0 \pm 0$ | $49 \pm 0.5$ | $51.4 \pm 0.3$ | $53.8 \pm 0.4$ | $53.8 \pm 0.4$ |
| -random-medium-expert | $34.4 \pm 2.6$ | $0 \pm 0.1$ | $22.9 \pm 3.3$ | $42.7 \pm 0.9$ | $46 \pm 1$ | $47.3 \pm 0.7$ |
| **DogTrot** | | | | | | |
| -expert | $98 \pm 0.7$ | $0.1 \pm 0.1$ | $94.4 \pm 0.5$ | $99.5 \pm 0.7$ | $98.9 \pm 2.2$ | $101.2 \pm 1.6$ |
| -medium-expert | $62 \pm 3.7$ | $0 \pm 0.3$ | $74.9 \pm 3.5$ | $84.8 \pm 3.7$ | $89.3 \pm 1.4$ | $93.9 \pm 1.4$ |
| -medium | $43.8 \pm 0.5$ | $0.1 \pm 0.1$ | $49.1 \pm 0.6$ | $46.5 \pm 0.5$ | $52 \pm 0.3$ | $50.2 \pm 0.2$ |
| -random-medium-expert | $37.2 \pm 3.6$ | $0.1 \pm 0.2$ | $0.1 \pm 0.1$ | $43.4 \pm 0.5$ | $44.1 \pm 1.2$ | $44.9 \pm 0.7$ |
| **Sum** | | | | | | |
| -expert | $580.2 \pm 19.5$ | $4 \pm 8.8$ | $635.5 \pm 6.1$ | $660.2 \pm 19.9$ | $669.1 \pm 16.9$ | $647.8 \pm 13.7$ |
| -medium-expert | $368.6 \pm 44.5$ | $9.7 \pm 9.1$ | $544.6 \pm 15.7$ | $642.1 \pm 21.6$ | $620.2 \pm 17.6$ | $639.7 \pm 13.6$ |
| -medium | $248.9 \pm 20.5$ | $6.9 \pm 8.6$ | $290.1 \pm 20.1$ | $315.7 \pm 15.6$ | $307.2 \pm 14.9$ | $320.6 \pm 16.7$ |
| -random-medium-expert | $172.9 \pm 24.7$ | $51.2 \pm 12.4$ | $88.6 \pm 34.5$ | $394.5 \pm 21.5$ | $331.5 \pm 24$ | $353.2 \pm 23.5$ |
| -all | $1370.6 \pm 109.2$ | $71.8 \pm 38.9$ | $1558.8 \pm 76.4$ | $2012.5 \pm 78.6$ | $1928 \pm 73.4$ | $1961.3 \pm 67.5$ |

## D  Decomposition comparisons

In this Section we compare the DecQN decomposition to two alternative methods that can be used for factorisable discrete action spaces. The first is based on the Branching Dueling Q-Network (BDQ) proposed by Tavakoli et al. (2018). Using our notation, each utility function is considered its own independent Q-function, *i.e.*

$$Q_{\theta_i}^i(s, a_i) = U_{\theta_i}^i(s, a_i) \,. \tag{5}$$

Each Q-function is trained by bootstrapping from its own target, and no decomposition is used. That is, the target for $Q_{\theta_i}^i(s, a_i)$ is given by $y = r + \gamma \max_{a_i' \in \mathcal{A}_i} Q_{\bar{\theta}_i}^i(s', a_i')$. The findings of Ireland & Montana (2023)

Table 7: Individual performance comparison $n = \{3, 10, 30, 50, 75, 100\}$. Figures are mean normalised scores $\pm$ one standard error, with 0 and 100 representing random and expert policies, respectively. At the bottom of the table we also provide totals, split by data quality and the entire benchmark. We see across all datasets that DecQN-IQL/OneStep perform best, followed by DecQN-CQL and DecQN-BCQ.

| Environment -dataset | BC | DecQN | DecQN-BCQ | DecQN-CQL | DecQN-IQL | DecQN-OneStep |
|---|---|---|---|---|---|---|
| DogTrot ($n = 3$) | | | | | | |
| -expert | $98 \pm 0.7$ | $0.1 \pm 0.1$ | $94.4 \pm 0.5$ | $99.5 \pm 0.7$ | $98.9 \pm 2.2$ | $101.2 \pm 1.6$ |
| -medium-expert | $62 \pm 3.7$ | $0 \pm 0.3$ | $74.9 \pm 3.5$ | $84.8 \pm 3.7$ | $89.3 \pm 1.4$ | $93.9 \pm 1.4$ |
| -medium | $43.8 \pm 0.5$ | $0.1 \pm 0.1$ | $49.1 \pm 0.6$ | $46.5 \pm 0.5$ | $52 \pm 0.3$ | $50.2 \pm 0.2$ |
| -random-medium-expert | $37.2 \pm 3.6$ | $0.1 \pm 0.2$ | $0.1 \pm 0.1$ | $43.4 \pm 0.5$ | $44.1 \pm 1.2$ | $44.9 \pm 0.7$ |
| DogTrot ($n = 10$) | | | | | | |
| -expert | $97.3 \pm 1.7$ | $-0.3 \pm 0$ | $96.8 \pm 1.8$ | $99.2 \pm 1.3$ | $106.5 \pm 1.1$ | $113.4 \pm 0.9$ |
| -medium-expert | $58.1 \pm 5.5$ | $0.5 \pm 0.1$ | $82.3 \pm 4.6$ | $83.4 \pm 2.1$ | $105.1 \pm 2.1$ | $109.8 \pm 1.2$ |
| -medium | $34.7 \pm 0.4$ | $0.5 \pm 0.2$ | $36.9 \pm 0.3$ | $39.1 \pm 0.6$ | $41 \pm 0.8$ | $47.2 \pm 0.4$ |
| -random-medium-expert | $33.6 \pm 4.4$ | $0.1 \pm 0.1$ | $6.1 \pm 2.8$ | $33.4 \pm 1.5$ | $52.1 \pm 2.3$ | $45.4 \pm 4.1$ |
| DogTrot ($n = 30$) | | | | | | |
| -expert | $99.7 \pm 0.4$ | $-0.3 \pm 0$ | $98.5 \pm 1$ | $99.8 \pm 0.8$ | $102.8 \pm 0.4$ | $106.4 \pm 0.8$ |
| -medium-expert | $70.3 \pm 4.9$ | $0.6 \pm 0.2$ | $73.2 \pm 8.7$ | $90.7 \pm 3.7$ | $98.3 \pm 1.6$ | $100.7 \pm 3$ |
| -medium | $33.1 \pm 0.2$ | $0.7 \pm 0.2$ | $29.1 \pm 0.5$ | $33.4 \pm 0.4$ | $39.9 \pm 0.1$ | $40.5 \pm 0.5$ |
| -random-medium-expert | $22.4 \pm 0.9$ | $0.2 \pm 0.1$ | $6.1 \pm 0.8$ | $22.9 \pm 2.5$ | $31.6 \pm 1.3$ | $30.4 \pm 1.8$ |
| DogTrot ($n = 50$) | | | | | | |
| -expert | $98.2 \pm 0.5$ | $0.5 \pm 0.1$ | $98.6 \pm 0.6$ | $97.7 \pm 0.6$ | $99.2 \pm 0.9$ | $100.4 \pm 1.4$ |
| -medium-expert | $67.7 \pm 5$ | $-0.3 \pm 0$ | $68.7 \pm 4.2$ | $88.9 \pm 2.1$ | $95 \pm 2.1$ | $97.1 \pm 1.4$ |
| -medium | $34.9 \pm 0.4$ | $-0.3 \pm 0$ | $36.5 \pm 0.7$ | $36.7 \pm 0.2$ | $45.1 \pm 0.6$ | $43.1 \pm 0.7$ |
| -random-medium-expert | $21 \pm 2$ | $0.7 \pm 0.1$ | $9.3 \pm 1.5$ | $27.6 \pm 1.3$ | $27.9 \pm 2.4$ | $36.4 \pm 1.8$ |
| DogTrot ($n = 75$) | | | | | | |
| -expert | $98.4 \pm 0.8$ | $0.6 \pm 0.2$ | $100.2 \pm 0.6$ | $98 \pm 1$ | $101.4 \pm 1.4$ | $102.9 \pm 0.9$ |
| -medium-expert | $75.3 \pm 2.6$ | $0 \pm 0$ | $55.4 \pm 3$ | $85 \pm 2$ | $89.9 \pm 0.7$ | $97.9 \pm 2.5$ |
| -medium | $41.4 \pm 0.2$ | $0.5 \pm 0.1$ | $41.8 \pm 0.1$ | $42.5 \pm 0.1$ | $48.5 \pm 0.4$ | $47.8 \pm 0.5$ |
| -random-medium-expert | $27.7 \pm 0.7$ | $0.7 \pm 0.2$ | $4 \pm 0.8$ | $33.4 \pm 2$ | $45.3 \pm 2.9$ | $43.6 \pm 2.6$ |
| DogTrot ($n = 100$) | | | | | | |
| -expert | $96.4 \pm 1.1$ | $0.1 \pm 0$ | $95 \pm 1.8$ | $100.4 \pm 0.7$ | $103.2 \pm 0.4$ | $105.5 \pm 0.8$ |
| -medium-expert | $70.9 \pm 6.5$ | $0.4 \pm 0.1$ | $74.6 \pm 5.8$ | $74.4 \pm 5.1$ | $93.9 \pm 1$ | $101.5 \pm 2.5$ |
| -medium | $34.7 \pm 0.8$ | $0.1 \pm 0.1$ | $34.5 \pm 0.5$ | $34.8 \pm 0.7$ | $40.9 \pm 0.2$ | $41.5 \pm 0.5$ |
| -random-medium-expert | $18.9 \pm 2.7$ | $0.3 \pm 0.1$ | $0 \pm 0$ | $12.7 \pm 2.4$ | $24 \pm 2.8$ | $26.2 \pm 1$ |
| Sum | | | | | | |
| -expert | $588 \pm 5.2$ | $0.7 \pm 0.4$ | $583.5 \pm 6.3$ | $594.6 \pm 5.1$ | $612 \pm 6.4$ | $629.8 \pm 6.4$ |
| -medium-expert | $404.3 \pm 28.2$ | $1.2 \pm 0.7$ | $429.1 \pm 29.8$ | $507.2 \pm 18.7$ | $571.5 \pm 8.9$ | $600.9 \pm 12$ |
| -medium | $222.6 \pm 2.5$ | $1.6 \pm 0.7$ | $227.9 \pm 2.7$ | $233 \pm 2.5$ | $267.4 \pm 2.4$ | $270.3 \pm 2.8$ |
| -random-medium-expert | $160.8 \pm 14.3$ | $2.1 \pm 0.8$ | $25.6 \pm 6$ | $173.4 \pm 10.2$ | $225 \pm 12.9$ | $226.9 \pm 12$ |
| -all | $1375.7 \pm 50.2$ | $5.6 \pm 2.6$ | $1266.1 \pm 44.8$ | $1508.2 \pm 36.5$ | $1675.9 \pm 30.6$ | $1727.9 \pm 33.2$ |

demonstrate that, in the online setting, BDQ is unable to match the performance of DecQN. This is likely caused by the fact that, as each sub-action space is now learnt independently, the effects of other sub-actions are treated as effects of the environment dynamics. Due to the fact that each agent is continually updating its own policy, this leads to non-stationary environment dynamics, making the learning problem much more challenging.

We also consider an alternative value-decomposition technique to the mean, namely the sum. That is, we replace the mean operator in Equation 2 with the sum operator:

$$Q_\theta(s, \mathbf{a}) = \sum_{i=1}^{N} U_{\theta_i}^i(s, a_i) .\tag{6}$$

Table 8: Individual performance comparison for DecQN-CQL using mean and sum decompositions and BDQ-CQL for $n = 3$. Figures are mean normalised scores $\pm$ one standard error, with 0 and 100 representing random and expert policies, respectively.

| Environment -dataset | DecQN-CQL (Mean) | DecQN-CQL (Sum) | BDQ-CQL |
|---|---|---|---|
| HumanoidStand | | | |
| -expert | $109 \pm 1.9$ | $95.1 \pm 1.8$ | $105.5 \pm 0.7$ |
| -medium-expert | $104.7 \pm 1.9$ | $86.1 \pm 2.8$ | $94.0 \pm 5.8$ |
| -medium | $51.4 \pm 0.3$ | $44.6 \pm 0.8$ | $48.2 \pm 0.5$ |
| -random-medium-expert | $42.7 \pm 0.9$ | $36.3 \pm 1.6$ | $43.2 \pm 0.5$ |
| DogTrot | | | |
| -expert | $99.5 \pm 0.7$ | $93.1 \pm 1.3$ | $94.9 \pm 1.2$ |
| -medium-expert | $84.8 \pm 3.7$ | $81.4 \pm 2.8$ | $75.7 \pm 3.6$ |
| -medium | $46.5 \pm 0.5$ | $44.3 \pm 0.3$ | $48.5 \pm 0.5$ |
| -random-medium-expert | $43.4 \pm 0.5$ | $39.9 \pm 0.9$ | $37.9 \pm 2.3$ |

Whilst this may seem a subtle change, Ireland & Montana (2023) proved that the mean and variance of the learning target under this decomposition are both higher than DecQN. Empirical experiments by Seyde et al. (2022); Ireland & Montana (2023) also confirm the inferior performance of the sum decomposition compared to the mean.

In Table 8 we can see that the sum decomposition is less performant in each of the tasks and datasets than the mean. For BDQ, we see that whilst in some cases performance is better than using the sum decomposition, it is generally still less performant than using the mean decomposition. Owing to these results, we focus on the mean decomposition in our main work.

