# OpenReview forum: "Benchmarking Offline Reinforcement Learning in Factorisable Action Spaces"
_TMLR — Rejected by TMLR_

### Review · Reviewer_Njxx · 2024-05-07

**Summary Of Contributions:**

This paper studies the problem of offline RL in environment where the action space is factorisable. A factorisable action space is a large action space where actions live in a cross-product of smaller action spaces.

This works is centered around three main contributions:

- theoretical insights showing why factorize actions would have benefits over atomic actions
- the introduction and benchmark of four algorithms designed for this problem.[ The four algorithms are combinations of DecDQN -- an online algorithm designed for factorized actions --, and of a known offline RL algorithm (CQL, IQL, ...)
- the introduction of environment to benchmark this problem, similar to what was done in d4rl.

**Audience:**

Yes

**Broader Impact Concerns:**

No concern.

**Claims And Evidence:**

No

**Requested Changes:**

I suggest a few changes:

1. please adress the comments in the minor weakness section

2. I would suggest to remove section4, and add a paragraph in the related work section mentioning theoretical works on overestimation and decDQN.

**Strengths And Weaknesses:**

# Strengths

## Algorithms

The proposed algorithms are straightforward and natural baseline, as they combine a simple algorithm for factorized actions with known and popular algorithms for offline RL. I think it is a good contribution to the community to know how these methods behave in practice. This work proposes quite a large benchmark of methods.


## Environments

Adding new datasets, with new specificities, is a very useful contribution.



# Weakness

This part is organized in two sections. The "main weakness" is the reason why I currently select "no" for claim and evidence. The "other weaknesses" part list minor and easier to fix issues.


## Main weakness: theory

This part is about section four.

In short, I think lemma 1 is false (or at best not informative) and that it compromises the rest of the section. Let me explain.

The main issue for lemma one is the way the error is defined. In section 4 the error is defined as the difference between $Q_\theta$ and $Q^\pi$, where $\pi$ is not defined but I will assume $Q^\pi$ is the q-value the algorithm is optimizing for.

Here, the problem is that you cannot state that $\epsilon = Q_\theta - Q^\pi$  is random, centered and uniformly distributed: this quantity is the error resulting from q-learning algorithm, not an error term like an approximation error for example. A convincing point is that, if $Q_\theta - Q^\pi$ is already a centered quantity, then we don't have to do any learning since $Q_\theta$ is already an estimator of $Q^\pi$. This is very important, because the definition of $\epsilon$ is the basis for Lemma 1 and Theorem 1, which I thus think are false.

For reference on how to properly define noise in a q-learning setup, consider the work of (Thrun & Schwartz, 1993): here, the noise is injected within the q-learning update, where it is reasonable to assume it is random and uniform, but it changes the behavior of the errors in the long run (see the definition of $Z_s$ in  (Thrun & Schwartz, 1993)).

To complement this, in (Ireland and Montana, 2024), the authors actually prove the opposite of the result of Lemma 1. Indeed, Theorem 1 in (Ireland and Montana, 2024) proves that the variance of the error of DecDQN is **higher** than the variance of DQN, which contracticts lemma 1 in this work.

That said, I think the paper would still be a sufficient contribution with section 4 entirely removed, and a paragraph in the related works section to mention the works above.

## Minor weaknesses

### Environments

There are missing details as how the factorization of actions is done in the environments.

Specifically:
 - what is the difference between the method in this paper versus factorizing actions in d4rl directly ?

### Algorithms

It would be nice to have an aggregate score of all algorithms,to maybe see if one algorithm has an statistically significant edge.


## Typos
In the proof of Lemma 1, there is a $(b − b)^2 = b^2$.

---

> ### Author Response · Authors · 2024-06-10
> **Response to review**
>
> Thank you once again for providing valuable feedback on our work and for allowing us the extra time to provide what we hope is a satisfactory response to the points you raised.
>
> ## Strengths
>
> We are pleased you feel our contribution is of benefit to the research community and that you recognise the importance of studying how offline-RL methods behave in settings where action spaces are factorisable.
>
> ## Weaknesses and requested changes
>
> ### Theory
>
> Upon reflection, we agree with your assessment of our theoretical analysis.   Rather than focus on the pre-update errors in Q-value estimates from function approximation as we have done, it is more informative to consider the post-update error as a result of the q-learning algorithm.  This is the approach taken by (Thrun & Schwartz, 1993) and (Ireland & Montana, 2024), where the target difference is treated as a random variable and its expectation and variance assessed.  We have since attempted to reframe our analysis is a similar manner, incorporating our notion of in-distribution and out of distribution errors into the target difference.  This has proven to be more complex than in the case of a single error distribution, as the maximisation operation is now over a pooled sampled from two different distributions (errors sampled from $\epsilon^{in}$ and $\epsilon^{out}$).  Unfortunately, due to personal circumstances, we have been unable to complete this analysis in the time allotted for revisions.  As you have indicated our contribution remains sufficient in the absence of this analysis, at your suggestion we have removed this Section from the paper and added the above works to Related Works.  We intend to continue with this analysis as part of future work.
>
> ### Clarity on factorisation process
>
> As per your request, we have added further details in Section 6 regarding the factorisation process.  Specifically in response to your query regarding the D4RL dataset, our set-up is based on a discretisation of continuous actions whereas D4RL retains the continuous nature of the actions.  When collecting our datasets, agents can only interact with the environment using a subset of actions determined by the discretisation process whereas the D4RL datasets are collected by agents with no restrictions on actions.  It is not possible to directly discretise the D4RL datasets as this would lead to an erroneous rewards/next states because the discretised actions would not correspond to the true continuous actions chosen (unless each dimension of the continuous action lies on a boundary for discretisation).  One could introduce discretised datasets based on the environments used in D4RL but we opted to use the DMC suite as it offers a broader range of tasks and has been used in previous work based on value-decomposition (Seyde, 2022)  (Ireland & Montana, 2024).
>
> ### Aggregated scores
>
> As per your request, to better compare performance across the entire benchmark we have added aggregated scores to our two main results tables in the Appendix (n=3 for all datasets and n=3, 10, 30, 50, 100 for dog-trot datasets).  We have done this split by data quality (i.e. expert, medium etc.) as well as the entire benchmark.  We have added the following comments to each table caption:
>
> - (For the n=3 datasets) - We see that DecQN-BCQ is the least performant of the offline methods and that DecQN-CQL/IQL/OneStep perform similarly overall and on expert, medium-exert and and medium datasets and DecQN-CQL the best on random-medium-expert datasets.
> - (For the n=3, 10, 30, 50, 100 dog-trot datasets) - We see across all datasets that DecQN-IQL/OneStep perform best, followed by DecQN-CQL and DecQN-BCQ.

---

### Review · Reviewer_ikZ7 · 2024-05-09

**Summary Of Contributions:**

This work investigates offline reinforcement learning in factorizable action spaces, where global actions consist of multiple distinct sub-actions. Inspired by prior work on online-RL, the authors advocate value-decomposition approaches in the spirit of the Decoupled Q-Network framework (DecQN), where the Q-value is approximated by a mean of several utility functions, one per sub-action space. The idea is that the value estimates for each sub-action space are computed independently, while the Q-value update considers their combined impact . As a result, the total number of actions for a which the Q-value must be learnt is substantially lower than that of doing learning on the joint action space. The learning process is also significantly simplified, which can result in better performance despite the additive approximation of the Q-value decomposition. The authors quantify the benefits theoretically, and argue that the proposed approach does not result in bias, while being able to reduce the variance, most notably for out-of-distribution actions as a result of higher sub-action coverage.

The authors discuss how to apply Dec-QN with factorizable action spaces to several existing offline-RL algorithms: Batch Constrained Q-learning (DecQN-BCQ), Conservative Q-learning (DecQN-CQL), Implicit Q-Learning (DecQN-IQL), and one-step approaches (DecQN-IQL). Given no benchmarks exist for factorizable action spaces, the authors introduce their own benchmarks by discretizing various environments/tasks from the DeepMind control suite. For the training data, they train agents to expert and medium performance levels, or even use random transitions, to create "expert", "medium", "medium-expert" and "random-medium-expert" datasets). The paper shows: (i) DecQN offline-RL vastly outperforms Atomic-CQL, in terms of performance and execution time; (ii) DECQN offline-RL methods consistently outperform behavioral cloning (except for DecQN-BCQ), and are able to extract near-expert policies (except for the challenging "random-medium-expert" dataset; and (iii) the developed DecQN methods are robust to increases in bin size per action subspace.

**Audience:**

Yes

**Broader Impact Concerns:**

I do not have broader impact concerns.

**Claims And Evidence:**

No

**Requested Changes:**

I consider all requested changes critical. That said, the authors may decide to address only some of them, as long as they adequately explain why they do not address the rest of them as part of their rebuttal.

- The authors should fix the various typos.
- Some references are not up-to-date. For instance, (Seyde et al., 2022) was accepted at ICLR 2023, but the authors have still kept the old reference. The authors should do through all their references, and ensure they cite the up=to-date (preferably accepted) version of the various references.
- In DecQN-BCQ, page 7, the authors write $y=r(s,a)+\gamma/N\sum_{i=1}^N\max_{a_i : \rho^i(a_i)\geq\tau}U^i_{\theta_i}(s',a_i')$. It should be $y=r+\gamma/N\sum_{i=1}^N\max_{a_i' : \rho^i(a_i')\geq\tau}U^i_{\theta_i}(s',a_i')$,  i.e., use the next action $a_i'$ instead of the current action $a_i$.
- In DecQN-CQL, Equation (5) looks correct to me. However, I feel it would have made sense for the authors to justify why the sum $\frac{1}{N}\sum_{i=1}^N$ is placed outside of $\log\sum_{a_j\in\mathcal{A}_i} \exp(\cdot)$.
One might have been tempted to put the average inside the logarithm of the sum, so some brief clarification might have been handy there.
- The authors should clarify their theory assumptions and, in particular, the symbol $U_i^{\pi_i}(s,a_i)$. If the authors implicitly assume there that the true Q-function can be additively decomposed as the average over all $U_i^{\pi_i}(s,a_i)$, then the authors should explain why they feel that this is a valid and not overly restrictive assumption. Indeed, the whole purpose DecQN is to rewrite the function approximation as an average of utility function, but not necessarily the true Q-function itself. The authors may also want to confirm that the 0 bias only seems to hold if the true Q-function decomposes additively, which, as mentioned, is a very restrictive assumption.
- I believe the authors should discuss the limitations of their benchmarks, given these benchmarks only consist in discretizing each action variable through a fixed number of bins (all action variables by the same number of bins). What are the implications of such a choice, and what are the difficulties of developing a more realistic benchmark, e..g., where multiple action variables can be grouped in the same action sub-space?
- The only competitors for the DecQN offline methods that the paper introduces are Behavioral Cloning and the Atomic Variant. However, as previously discussed, the authors do not actually compare how their methods perform on their discretized environments compared to other offline RL methods on the original environments. If the authors could show that their methods can indeed yield a benefit, this would be a good sign that the proposed framework can indeed positively impact offline RL. If not, then at the very least the authors should discuss why this is the case while being transparent about the limitations of their framework for tackling existing offline RL benchmarks (after discretization).

**Strengths And Weaknesses:**

Strengths
- To the best of my knowledge, this is the first work to consider factorizable action spaces for offline RL. DecQN is a well-studied approach for traditional online RL with various benefits (e.g., Seyde et al., 2022). It is thus only natural for the authors to investigate this technique to the more challenging offline-RL setting.
- The authors explain how existing popular techniques in offline RL can be adapted to incorporate Dec-QN. This is mostly straightforward, but it is helpful to have the pseudocode for the adapted algorithms.
- The authors conduct a quite extensive experimental study, which demonstrates how factorized action sub-spaces with offline RL can outperform atomic offline RL, while they benchmark the various DecQN offline RL methods they propose. Furthermore, they are able to demonstrate that their algorithms are quite robust to larger bin sizes, which is important for scalability.
- The paper develops some theory to justify why the proposed framework is able to reduce the variance, most notably for out-of-distribution actions as a result of higher sub-action coverage. The theory on in-distribution vs. out-of-distribution actions seems an interesting contribution of this work.
- The created "expert", "medium", "medium-expert" and "random-medium-expert" datasets are a nice contribution, and important for the task of offline RL.
- The paper is not hard to follow, even though there are quite a few typos.

Weaknesses
- The paper novelty is limited. Essentially, the authors take the DecQN framework by Seyde et al. (2022) and apply the value decomposition ideas to the setting of offline RL. Even the algorithms that they explore are all based on existing offline RL methods. A notable exception is the theory related to our-of-distribution vs in-distribution actions, which predominantly concerns the offline RL setting.
- Part of the theory section seems to rely on some hard-to-justify assumptions. I was particularly worried about Equation (4), where the authors introduce the symbol $U_i^{\pi_i}(s,a_i)$. First, notice that this symbol was not introduce earlier, so it is not immediately clear what the authors might mean by it. Now, if what they try to say is that $Q^{\pi}(s,a)=\frac{1}{N}\sum_{i=1}^N U_i^{\pi_i}(s,a_i)$, then this would imply that the true Q-function decomposes additively among the different agents/action-subspaces. However, this is clearly not the assumption in (Seyde et al., 2022). Indeed, the idea there is that the Q-function $\textit{approximation}$ can be additively decomposed among the various action subspaces. The original Q-function most probably does not perfectly decompose additively, and this would indeed be a very strict constraint. It is enough to simply assume that the function approximation decomposes additively. In fact, (Rashid et al., 2018) notice that "Strictly speaking, each $Q_a$ is a utility function (Guestrin et al., 2002) and not a value function since by itself it does not estimate an expected return."
For the same reason, I am somehow doubtful that the notation $\epsilon_{(s,a_i)}=U^i_{\\theta_i}(s,a_i)-U_i^{\pi_i}(s,a_i)$ is meaningful, since $U_i^{\pi_i}(s,a_i)$ may not even exist in the first place, when the true Q-function does not decompose additively.
I understand that the authors may have made this assumption for the purpose of developing some theory, but I feel that this is still an unreasonable assumption. Also, notice that if this assumption is not correct, then I do not think the authors can claim that DecQN has 0 bias, as they currently claim.
- In the absence of benchmarks for factorizable actions spaces, the authors claim that they create their our own set of benchmarks based on a discretized variant of the DeepMind control suite. However, unless I misunderstood the datasets, what the authors essentially do is to discretize each action variable using a fixed number of bins (e.g., $n_i=3,4,5,6$ etc.). Even though this is technically correct, it seems a rather straightforward thing to do. To me, it would have been much more interesting to create for instance action subspaces by grouping/clustering a small number of action variables. I feel that the current benchmarks are quite simple, and they cannot really account for the more general factorizable action space setting, where each action subspace can consist of several action variables. Also, I was unclear why all action variables had to be discretized the same number of bins $n_i$. In reality, shouldn't we expect that some action subspaces will contain a different number of bins than others?
- The authors can indeed show that DecQN performs better than the atomic variant and is also faster, especially as the bin size increases. This is quite intuitive and aligned with prior work on online RL. However, I am not convinced that this is the right comparison. Personally, I would have preferred to see whether on each of the environments, DecQN with offline RL on the discretized environment could outperform other offline RL methods that do not rely on discretization. Let me explain why. The authors introduce a framework in the hope that it can accelerate offline RL and improve performance. In this direction, they should provide results showing whether their proposed framework on the discretized environments can beat other offline methods on the original environments. Otherwise, their comparison cannot be considered conclusive. I understand the argument that no benchmarks exist, so the authors must create their own benchmarks. But if their competitor is by definition a weak algorithm, then it is hard to see whether factorizable action spaces with discretization can yield benefits over offline RL on the original environment without discretization. Given that the paper novelty is not significant, it would be at least nice for the authors to show that their framework can empirically beat other offline RL approaches (not just the atomic variant of behavioral cloning), instead of having a very narrow evaluation.
- There are quite a few typos. Some examples:

page 1: but offline => but in offline

page 2: where global actions consists => where global actions consist

pae 2: This benchmarks => This benchmark

page 9: minimsing the following loss => minimising the following loss

page 9: fullstop missing before sentence "The policy follows that of discrete-action advantage-weighted-behavioural-cloning..."

page 10: Fullstop missing from sentence "The full procedure can be found in Algorithm 4"

page 11: For each set of experiments we provide visual summaries with tabulated results available in the Appendix for completeness. => which Appendix exactly? Please specify.

page 11: a dramatic increases => a dramatic increases

page 12: these results supports our claim => these results support our claim

page 13: failing to learn to a policy => failing to learn a policy

page 24: lower quality datasets benefits => lower quality datasets benefit

page 27: we compare to two => we compare two

---

> ### Author Response · Authors · 2024-06-10
> **Response to review**
>
> Thank you once again for providing valuable feedback on our work and for allowing us the extra time to provide what we hope is a satisfactory response to the points you raised.
>
> ## Strengths
>
> We are pleased you recognise the importance of studying offline-RL in factorisable action spaces and our initial contribution in this area, particularly our empirical work.
>
> ## Weaknesses and requested changes
>
> ### Novelty
>
> Our main aim for this paper is to conduct an initial investigation into offline-RL in factorisable action spaces.  Rather than develop new algorithms specifically tailored to this setting, we instead want to provide a foundation for the research community to build upon.  Following the examples of D4RL and RL Unplugged, our aim is to create a consistent evaluation protocol through the use of benchmark datasets which can drive the development of new approaches.  Similar to D4RL and RL Unplugged, we use established algorithms as part of our initial evaluation, although in our case this involves some algorithmic development since existing offline-RL methods have yet to be adapted to factorisable action settings.
>
> ### Theory
>
> After reassessing our theory following feedback from reviewer Njxx, we have concluded an analysis focused on errors in Q-values estimates from function approximation resulting from the q-learning algorithm would be more informative.  This is the approach taken by (Thrun & Schwartz, 1993) and (Ireland & Montana, 2024), where the target difference is treated as a random variable and its expectation and variance assessed.  We have since attempted to reframe our analysis is a similar manner, incorporating our notion of in-distribution and out of distribution errors into the target difference.  This has proven to be more complex than in the case of a single error distribution, as the maximisation operation is now over a pooled sampled from two different distributions (errors sampled from $\epsilon^{in}$ and $\epsilon^{out}$).  Unfortunately, due to personal circumstances, we have been unable to complete this analysis in the time allotted for revisions.  As such, at the suggestion of reviewer Njxx, we have removed this Section from the paper and defer such analysis to future work.  However, despite removing this theory, we feel our extensive empirical evaluation still fulfils our objective of providing insights into offline-RL in factorisable action spaces and our benchmark remains a useful tool for research in this area.
>
> ### Benchmark dataset
>
> We take on board your comments about variable bin sizes $n_i$.  For the purpose of this initial investigation we used equal bin sizes out of practical considerations.  If using variable bin sizes, we’d have to consider what size for each action dimension, which opens up many possibilities and necessitates more careful planning/experimentation to ensure datasets aren’t redundant.  In the future, we hope to expand our benchmark datasets to incorporate features such as variable bin sizes, masked sub-actions and clustering of sub-action dimensions, as well as environments that are more representative of real-world scenarios involving factorisable actions spaces.  As per your request, we have noted the limitations of our benchmark and plans for these future additions in the revised draft.
>
> Regarding your comment about comparing to offline-RL methods in the original environments.  This likely stems from a lack of clarity on our part regarding the type of actions we are interested in.  In this work we are not concerned with continuous actions, our focus in purely on discrete actions.  We are not taking a dataset of continuous actions and then discretising them, rather our datasets are collected by agents that can only interact with the environment using a subset of actions determined by the discretisation process.  For example, if the continuous action range is originally [-1, 1], for n=3 our discrete actions become {-1, 0, 1}.  Despite using the DMC suite, our focus is not continuous control.  We use the DMC suite because actions from environments/tasks can be discretised such that their combinatorial action spaces becomes prohibitively large (see Table 1), plus we have command over the discretisation process allowing us to investigate specific aspects (see above re: variable bin sizes).  Finally, please note that the BC results are from a factorised BC policy not an atomic BC policy.  We have taken the opportunity to make all this clearer in the revised draft.
>
> ### Typos and references
>
> Finally, thank you for pointing out typos and inaccuracies with references, which we have corrected.

---

### Review · Reviewer_pBQN · 2024-05-13

**Summary Of Contributions:**

This works studies offline RL in factorisable action spaces, which is less studied in existing literature. Analogous to D4RL, this work establishes a offline dataset of varying quality and different tasks based on a discretised variant of the DeepMind Control Suite, which can serve as a testbed for offline RL methods in factorisable action spaces.

This work proposes several methods based on DecQN, by incorporating techniques and ideas in existing offline methods into the individual utility functions in DecQN. This work also provides theoretical results on demonstrating the superiority of decomposed value learning over atomic value learning in the variance of value estimation error under certain assumptions.

With proposed offline dataset and methods, this work presents a benchmark for offline RL methods in factorisable action spaces, also demonstrating the effectiveness of proposed methods.

**Audience:**

Yes

**Claims And Evidence:**

Yes

**Requested Changes:**

- Add discussions for the related works I mentioned above or justify the reason why some of them are not related.
- Discuss more about the independent noise assumption and the reasonability of discrete DMC in representing real-world problems with factorisable action spaces.
- I also expect to see some more efforts in taking a more practical environment into this work and I think it will make way more impact. This is not mandatory.

**Strengths And Weaknesses:**

###Strengths:

- The paper is clearly written and most details can be found in appendix.
- To my knowledge, the contributions of this work are new and missing in the literature. And I think the dataset and proposed methods (along with the code) will be useful.
- Although the theoretical results are simple and the proposed methods are straightforward, they are intuitive and make sense. Also, limitations are discussed in this paper, pointing out some interesting future directions.

###Weaknesses:

- Related works on addressing factorisable action space/large action space in online or offline learning setting [1-7] should be discussed.
- The dataset established by discretizing continuous DMC cannot fully convince me. I think it will be a useful and convenient dataset to evaluate algorithms and methods, but I remain a concern on its reasonability to represent real-world problems with factorisable action spaces. I think a problem like recommendation system used in [5] makes more sense to me.
- The theoretical results are proposed based on the assumption of independent uniform noise for each sub-action. It is intuitive but quite limit. I think more discussion on it is needed.
- The proposed methods are based on DecQN. It uses a linearly additive relationship between sub-actions, like VDN in MARL literature. While more complex and expressive relationships like QMIX or other ones that obey IQM principle are expected. I also see that the authors discuss on this point in the last section.

---
Reference:

[1] Deep Reinforcement Learning in Large Discrete Action Spaces. https://arxiv.org/abs/1512.07679

[2] Q-Learning in enormous action spaces via amortized approximate maximization. https://arxiv.org/abs/2001.08116

[3] Learning and Planning in Complex Action Spaces. ICML 2021

[4] Growing Action Spaces. ICML 2020

[5] Learning Pseudometric-based Action Representations for Offline Reinforcement Learning. ICML 2022

[6] HyAR: Addressing Discrete-Continuous Action Reinforcement Learning via Hybrid Action Representation. ICLR 2022

[7] PLAS: Latent Action Space for Offline Reinforcement Learning. CoRL 2020

---

> ### Author Response · Authors · 2024-06-10
> **Response to review**
>
> Thank you once again for providing valuable feedback on our work and for allowing us the extra time to provide what we hope is a satisfactory response to the points you raised.
>
> ## Strengths
>
> We are pleased you found our paper easy to follow.  Also, that you recognise the importance of research into offline-RL in factorsiable action spaces and that our contribution in this domain is useful.
>
> ## Weaknesses and changes
>
> ### Missing literature
>
> We thank you for providing additional literature for us to review and incorporate into our Related Work.  As per your request, we have added these papers to the Related Work section, with the exception of [6, 7] as these are focused on environments with combined discrete and continuous actions [6] or continuous actions [7], whereas our work is focused solely on discrete actions.
>
> ### Choice of environments
>
> We use the DMC suite because actions from environments/tasks can be discretised such that their combinatorial action spaces become prohibitively large (see Table 1).  In addition, we have control over the discretisation process, which in this paper we used to control the bin sizes, but in future work can be used to investigate specific aspects such as variable bin sizes, masked actions, clustered actions, and so on.  The number of discrete actions for tasks in [5] is either 1000 or 4096, which although reasonable large, can still be tackled using DQN, whereas we wanted to investigate scenarios where DQN couldn’t be used.  Plus, although some of the environments in [5] can be factorised, we have less control over this.  For example, there are only two possible actions for each sub-action dimensions for the maze tasks.  We note in our Discussion/Conclusion that the development of environments specifically tailored to factorisable action spaces would be of great benefit, and this is something we hope to work on in the future.
>
> ### Theory
>
> After reassessing our theory following feedback from reviewer Njxx, we have concluded an analysis focused on errors in Q-values estimates from function approximation resulting from the q-learning algorithm would be more informative.  This is the approach taken by (Thrun & Schwartz, 1993) and (Ireland & Montana, 2024), where the target difference is treated as a random variable and its expectation and variance assessed.  We have since attempted to reframe our analysis is a similar manner, incorporating our notion of in-distribution and out of distribution errors into the target difference.  This has proven to be more complex than in the case of a single error distribution, as the maximisation operation is now over a pooled sampled from two different distributions (errors sampled from $\epsilon^{in}$ and $\epsilon^{out}$).  Unfortunately, due to personal circumstances, we have been unable to complete this analysis in the time allotted for revisions.  As such, at the suggestion of reviewer Njxx, we have removed this Section from the paper and defer such analysis to future work.  However, despite removing this theory, we feel our extensive empirical evaluation still fulfils our objective of providing insights into offline-RL in factorisable action spaces and our benchmark remains a useful tool for research in this area.
>
> ### More complex and expressive decompositions
>
> We take on board your comments about more sophisticated methods of decomposing the Q-function, e.g. QMIX.  As you mention, this is something we allude to in our Discussion/conclusion section and we would be interesting in exploring in future work.

---

### Decision · Action_Editor_FWcS · 2024-06-21

**Recommendation:** Reject

**Comment:**

The paper received some critical reviews. In particular, numerous problems on the technical clarity were identified. The authors did a great job responding and addressing some of them, enough to convince one reviewer to turn from initially negative to somewhat positive. However, some changes were not received in time for another reviewer to see them, and whose official recommendation remained critical. Another reviewer remains unconvinced despite the changes made.

The other problem is that reviewers were not convinced by the chosen domains, finding them inadequate to support the claims. I personally found this criticism to be slightly overstated, but the main point remains: discretizing a continuous domain is not the only way to obtain large action spaces (it's not even the most natural way). There are many domains whose actions structurally decompose into an action hierarchy. The reviewer gives one as an example and the authors counter by stating that 4096 is not large enough, which is highly debatable -- but still is missing the point. The one reference is just one example of how you can obtain large, discrete factorizable action spaces but there are many, many more. Also, it is a fair to expect the method to work on "small" actions spaces of size 4096, even if DQN can handle action spaces of that size, because that action space might naturally decompose into a hierarchy of choices that the method should exploit.

Here are specific quotes of comments in official recommendations (spanning across reviewers):

"Part of the theory section seems to rely on some hard-to-justify assumptions. One such example is Equation (4), where the authors introduce the symbol $U_i^{\pi_i}(s,a_i)$. If what they try to say is that $Q^{\pi}(s,a)=\frac{1}{N}\sum_{i=1}^N U_i^{\pi_i}(s,a_i)$, then this would imply that the true Q-function decomposes additively among the different agents/action-subspaces. However, this is clearly not the assumption in (Seyde et al., 2022). Indeed, the idea there is that the Q-function $\textit{approximation}$ can be additively decomposed among the various action subspaces. The original Q-function most probably does not perfectly decompose additively, and this would indeed be a very strict constraint. It is enough to simply assume that the function approximation decomposes additively. For the same reason, I am somehow doubtful that the notation $\epsilon_{(s,a_i)}=U^i_{\theta_i}(s,a_i)-U_i^{\pi_i}(s,a_i)$ is meaningful, since $U_i^{\pi_i}(s,a_i)$ may not even exist in the first place, when the true Q-function does not decompose additively."

"I feel that the current benchmarks are quite simple, and they cannot really account for the more general factorizable action space setting, where each action subspace can consist of several action variables."

"The authors can indeed show that DecQN performs better than the atomic variant and is also faster, especially as the bin size increases. This is quite intuitive and aligned with prior work on online RL. However, I am not convinced that this is the right comparison. Personally, I would have preferred to see whether on each of the environments, DecQN with offline RL on the discretized environment could outperform other offline RL methods that do not rely on discretization."

"My major concern lies in the proposed benchmark and the experimental evaluation of the proposed method. Although I agree that factorizing/discretizing the continual action space of DMC environments can be a useful and convenient way to mimic a natural factorized action space, I do not think it is sufficient and representative."

Unfortunately the authors were unable to submit some of the changes in the time given, but the paper is interesting and I believe can be fixed with these two main critcism addressed.  Hence, I encourage the authors to take this feedback into account and submit an improved version of the paper.

**Audience:**

The audience is a good fit for TMLR.

**Claims And Evidence:**

Two of the three reviewers chose "No" for this question in their official recommendations.

There are two main issues with the paper:
- Technical clairity, in particular the theoretical statements
- Experiments in domains that are too small or not representative

I will elaborate on these below, but ultimately these were the main two criteria that ultimately led to the decision.

**Resubmission Of Major Revision:**

The authors may consider submitting a major revision at a later time.